# Comparing the efficiency of open and enclosed filtration systems in environmental DNA quantification for fish and jellyfish

**Sayaka Takahashi** [1,2]*, **Masayuki K. Sakata**[3], **Toshifumi Minamoto**[3], **Reiji Masuda**[2]

**1** Faculty of Life and Environmental Science, Shimane University, Matsue, Shimane, Japan, **2** Maizuru Fisheries Research Station, Kyoto University, Nagahama, Maizuru, Kyoto, Japan, **3** Department of Human Environmental Science, Graduate School of Human Development and Environment, Kobe University, Kobe, Hyogo, Japan

* tsayaka@life.shimane-u.ac.jp

**Data Availability Statement:** All relevant data are within the manuscript and its Supporting Information files.

## Abstract

Water sampling and filtration of environmental DNA (eDNA) analysis have been performed by several different methods, and each method may yield a different species composition or eDNA concentration. Here, we investigated the eDNA of seawater samples directly collected by SCUBA to compare two widely used filtration methods: open filtration with a glass filter (GF/F) and enclosed filtration (Sterivex). We referred to biomass based on visual observation data collected simultaneously to clarify the difference between organism groups. Water samples were collected at two points in the Sea of Japan in May, September and December 2018. The respective samples were filtered through GF/F and Sterivex for eDNA extraction. We quantified the eDNA concentration of five fish and two cnidarian species by quantitative polymerase chain reaction (qPCR) using species-specific primers/probe sets. A strong correlation of eDNA concentration was obtained between GF/F and Sterivex; the intercepts and slopes of the linear regression lines were slightly different in fish and jellyfish. The amount of eDNA detected using the GF/F filtration method was higher than that detected using Sterivex when the eDNA concentration was high; the opposite trend was observed when the eDNA concentration was relatively low. The concentration of eDNA correlated with visually estimated biomass; eDNA concentration per biomass in jellyfish was approximately 700 times greater than that in fish. We conclude that GF/F provides an advantage in collecting a large amount of eDNA, whereas Sterivex offers superior eDNA sensitivity. Both filtration methods are effective in estimating the spatiotemporal biomass size of target marine species.

## Introduction

Environmental DNA (eDNA) analysis is attracting a great deal of attention as a more efficient and sensitive tool than conventional monitoring methods [1, 2]. After Ficetola [3] applied a newly developed eDNA method to detect bullfrogs in ponds, researchers have attempted to

**Funding:** This study was partly supported by the CREST program from the Japan Science and Technology Agency (grant number: JPMJCR13A2; http://www.jst.go.jp/kisoken/crest/en/project/33/e33_13.html) and JSPS Grant-in-Aid for Scientific Research (B) 19H03031, Japan. The funders had no role in study design, data collection and analysis, decision to publish, or preparation of the manuscript.

**Competing interests:** The authors have declared that no competing interests exist.

use it for detecting animals and plants, and for quantifying their abundance in the environment. Ocean field studies comparing the species-specific eDNA sequence method with conventional survey methods [4], or the eDNA metabarcoding analysis with the underwater visual census [5], bottom trawling [6] and net and trap [7] methods, suggest that the eDNA analysis is a promising tool for revealing the species composition of fish communities. By using the eDNA metabarcoding analysis, researchers managed to detect 93.3% of the fish species present in seawater samples from aquarium tanks [8].

For quantification using eDNA analysis, there was a positive correlation between eDNA concentration and fish biomass in ponds [9, 10] and tanks [11, 12]. The concentration of eDNA was also positively correlated with the size of fish [13], density [14–16] and wet mass [17]. Previous studies comparing spatiotemporal change in abundance or biomass in aquatic species, based on traditional methods and eDNA concentration, found a significantly positive correlation between biomass and the amount of eDNA in visual observations via land or vessel based surveys [18], using commercial fish landing data [19], via captures by bottom trawl [6], and by monitoring using echo sounder technology [20] in marine environments. A similar correlation was detected in a snorkeling survey [21], net capture surveys [9, 10], and during mark-recapture experiments [14] in freshwater environments; however, some researchers reported that a quantitative relationship between biomass and eDNA abundance was not found [22–24].

Water sampling, filtration, preservation and DNA extraction methods vary depending on the sample type and the research team conducting the eDNA analysis of species composition and quantification of aquatic organisms [25, 26]. Differences in filtration and sampling protocol affect the amount of eDNA detected [27–30] and the detection rate [31–34] of aquatic species, which is problematic. These studies elucidate the necessity to choose suitable filters, which vary depending on the target species, taking into account environmental factors and water sample types to establish an optimized and versatile protocol.

Recently, a simple, on-site eDNA analysis system was developed [11, 35]. For on-site filtration, an enclosed Sterivex filter is handier and more effective than an open filter; the latter requires a comparatively larger-scale filtration system. The transition from open filter (requiring handling, a filter funnel and a vacuum pump) to enclosed filter (enclosed in a capsule during filtration and DNA extraction) has advanced. Sterivex filtration is used as a method to examine the presence/absence of target species [36, 37]. Filtration time is shortened by combining the filter with a syringe, in situations where on-site filtration is required [38]. It has also been reported that eDNA is better conserved and a greater amount of it can be extracted when using an enclosed filter, as opposed to an open filter, to detect fish species in ponds [39]. This may be particularly true when using the eDNA metabarcoding method that utilizes MiFish PCR primers, whereby the number of species detected by using Sterivex filters, was significantly higher than the corresponding number obtained by using glass fiber filters (GF/F) [31]. On the other hand, the amount of eDNA obtained by open filtration was larger than that obtained by the precipitation method [27, 29]. When comparing open filtration systems, a greater amount of eDNA was extracted by using a cellulose nitrate filter [30] or by using a GF/F [27]; the most generally used pore sizes were 0.45 μm and 0.7 μm [26], the latter being used in this study. In the water, eDNA is considered to exist in various states and particle sizes [2], most abundantly in the 1–10 μm size class [40]. It is therefore efficient to use a 0.8 μm-pore size filter for filtration [38] and a 0.7 μm-pore size GF/F has been recommended for time and cost effectiveness [28], as shown in previous studies [18–21, 35, 41–43].

Some controversy exists as to whether there are any differences between the amount of eDNA detected by the two different filtration methods (GF/F and Sterivex) when performing quantitative analysis of eDNA. A direct comparison of eDNA concentrations obtained by the

two filtration methods will clarify the differences between them. Based on such knowledge, adaptive usage of each filtration method will be possible, and this should facilitate the ability to estimate the biomass of target species.

The present research aimed to test whether there is a correlation between the eDNA concentrations obtained by GF/F and Sterivex, and between the eDNA concentration and biomass. To check for a potential bias between the upper and lower layers of sampling bags, the eDNA concentrations of these two layers were also compared. Comparing the estimated biomass using different eDNA methodologies and applications can be valuable for future eDNA studies, particularly for optimizing survey strategies [25].

## Materials and methods

### Ethics statement

The underwater visual survey was conducted in accordance with local and governmental laws and regulations. Underwater surveys in Nagahama were approved by the harbormaster of Maizuru Bay (No. 300 issued on July 6 and No. 405 issued on September 28, 2018). No approval was required for the surveys in Otomi where leisure diving is common. No fish or other animals were harmed for the purpose of this study, except for tissue sampling for genetic analyses (see 'PCR analysis' below). The research (observation, fish collection, tissue sampling and euthanasia) was performed according to the guidelines of Regulation on Animal Experimentation at Kyoto University (https://www.kyoto-u.ac.jp/en/research/research-compliance-ethics/animal-experiments.html, last accessed on November 14, 2019) and the Kyoto Prefecture Fishery Management Rules (https://www.pref.kyoto.jp/reiki/reiki_honbun/aa30006341.html, last accessed on November 14, 2019). No ethical approval was required for this procedure due to the common consumption of these fishes. Our field studies did not involve endangered or protected species.

### Water sampling and filtration in Otomi

Seawater samples (3 L each) were collected using a water sampling bag, Lamizip (Standup Nylon Bag with Zipper, LZ-14, Seisannipponsha, Ltd., Tokyo, Japan) at approximately ten min intervals at 1 m off the bottom, at six locations of Otomi, Wakasa Bay, Fukui, Japan [44] (35˚ 32N, 135˚30E; Figs 1A, 1D and 2) on May 15, 2018. Water temperature (measured using an alcohol thermometer that had been calibrated by a mercury standard thermometer) was 20.2˚C near the sea surface and 18.2˚C at seafloor, salinity (measured by a water quality meter with a conductivity probe and reported in practical salinity units (psu): LAQUAact ES-71, Horiba, Ltd., Kyoto, Japan) was 30.4, visibility (visually estimated by Masuda) was 8 m, and depth (measured with a diving computer: SUUNTO D6, Vantaa, Finland) was 3.0–5.8 m.

The eDNA analysis process was conducted based on the methods previously described in the Environmental DNA Sampling and Experiment Manual (Version 2.1) [45], with a slight modification. One liter of each sample was poured into a prewashed plastic bottle, and 1 mL of benzalkonium chloride (BAC, Nihon Pharmaceutical, Co., Ltd., Tokyo, Japan) was added. Another 1 L of each sample was filtered through Sterivex (0.45 μm-pore size, Merck Millipore, Darmstadt, Germany) using a 50-mL syringe, and 2 mL RNA*later* (Thermo Fisher Scientific, Waltham, MA, USA) was added for DNA preservation. As a blank control, 500 mL purified water (Kenei Pharmaceutical, Osaka, Japan) was filtered in the same way using a measuring cup bleached with 0.1% sodium hypochlorite and washed with purified water. Bottles and Sterivex filters were transported on ice in a cooler box to the laboratory. It took less than 60 min from sampling to Sterivex filtration.

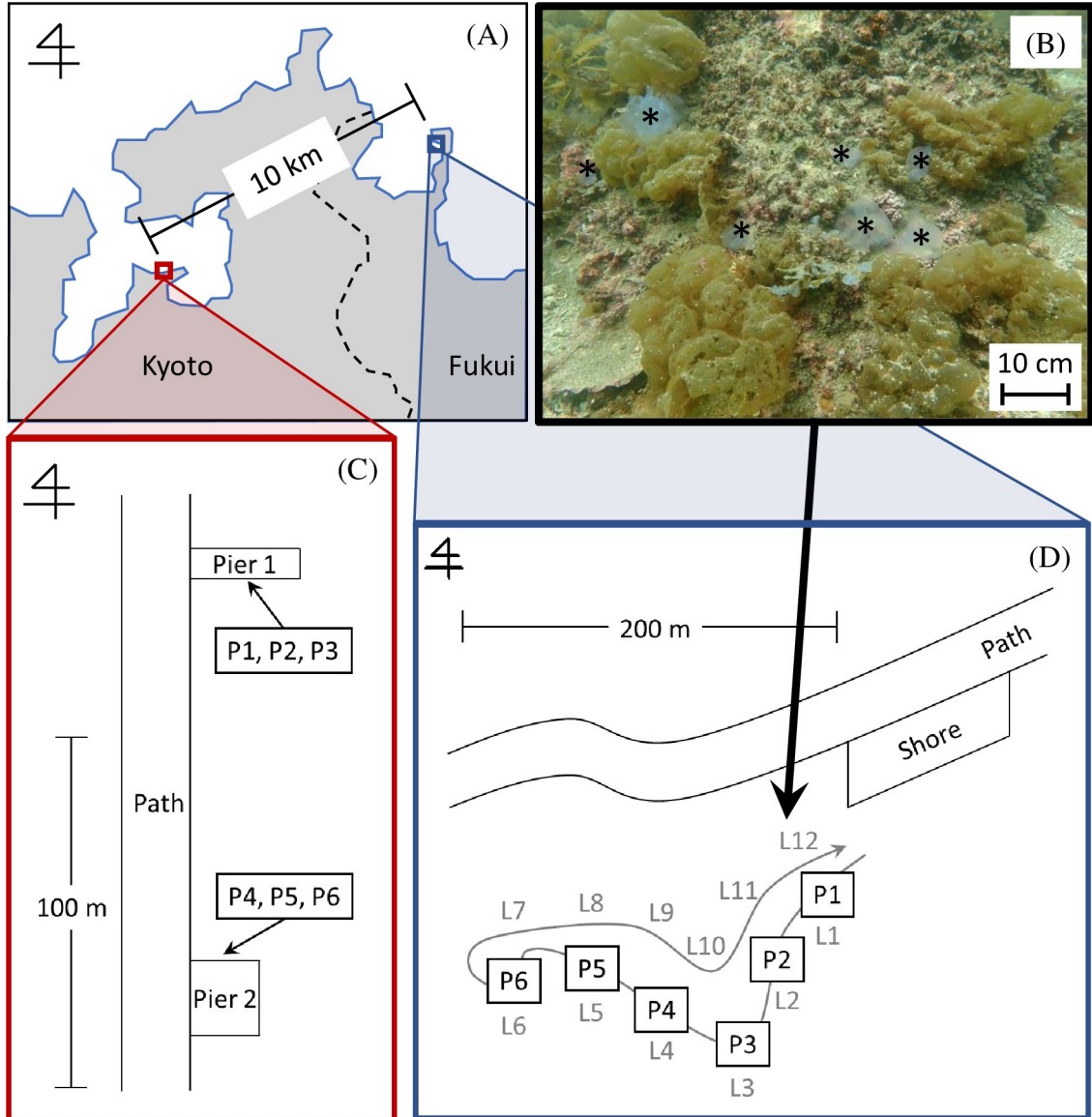

**Fig 1. Study site.** (A) Map showing the study sites in Kyoto and Fukui Prefecture. (B) Photograph of *Aurelia aurita* carcasses (*) on the sea floor found along the Line (L) 12 (black arrow) in Otomi. (C) Sample points (P1-P3 at pier 1 and P4-P6 at pier 2) in Nagahama. (D) Sampling points (P1-P6) in Otomi along the westward (outgoing, L1-L6) way of visual census lines (grey curved arrow). Visual census was also conducted along the eastward (return, L7-L12) way.

Each 1 L sample bottle and 500 mL of distilled water, was filtered through an aspirator using glass fiber filters (GF/F, 0.7 μm-pore size, Whatman, Maidstone, UK) in the laboratory. Filtering devices were bleached after every filtration with 0.1% sodium hypochlorite for 5 min, washed with tap water, and rinsed with distilled water. Filters were wrapped in aluminum foil, placed in plastic bags, and preserved at -20°C until DNA extraction. The process from sampling to preservation was conducted within seven hours. Nitrile gloves were worn both during filtration and the procedures that followed.

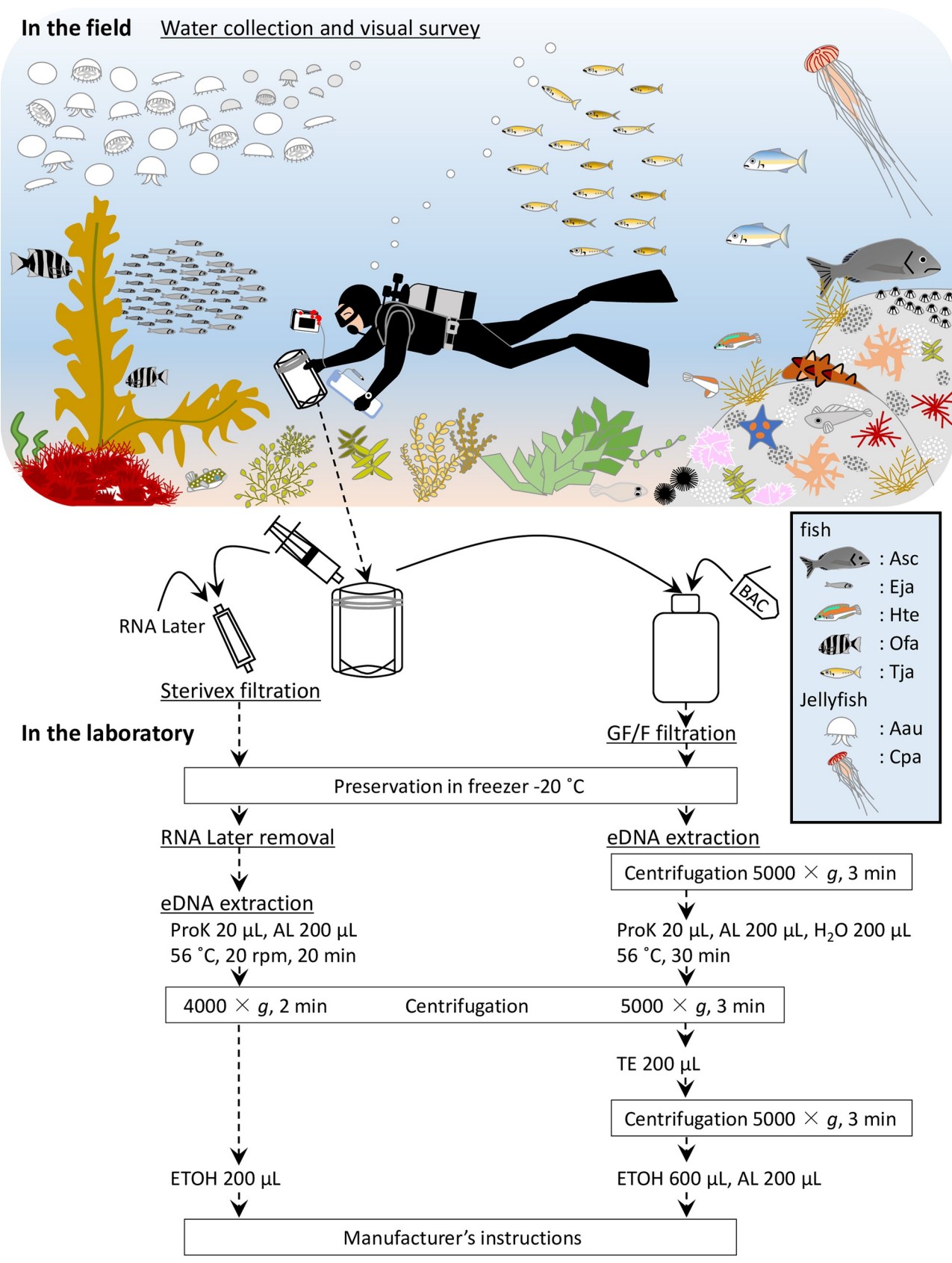

**Fig 2. Schematic drawing of visual survey, and outline flowchart of water collection, extraction, and comparison of filtration using GF/F and Sterivex.** The illustration depicts two species of jellyfish (Aau and Cpa), pelagic fish (Eja and Tja) and demersal fish (Ofa, Hte and Asc) and highlights a visual representation of the position in the water column and schooling behavior of some of these species assemblages during a typical transect. Seawater samples, collected during visual survey, were filtrated using Sterivex on site, while samples for GF/F filtration were transferred to the laboratory. BAC: benzalkonium chloride, ProK: Proteinase K, AL: buffer AL, TE: buffer TE, ETOH: ethanol, Asc: *Acanthopagrus schlegelii*, Eja: *Engraulis japonicus*, Hte: *Halichoeres tenuispinnis*, Ofa: *Oplegnathus fasciatus*, Tja: *Trachurus japonicus*, Aau: *Aurelia aurita*, Cpa: *Chrysaora pacifica*.

## Water sampling and filtration in Nagahama

Seawater samples (3 L each) were collected at 8–10 min intervals at 1 m off the bottom at six locations (three at each of two piers in the Maizuru Fisheries Research Station of Kyoto University) [46] (Nagahama, Maizuru, Kyoto, Japan; 35°29N, 135°22E; Figs 1A, 1C and 2) on September 19, 2018. Environmental data collection and the process from water sampling to preservation was performed in the same manner as at the Otomi site. The water temperature, salinity and depth near pier 1 and pier 2 was 26.4°C, 30.0 and 3.7 m, and 26.2°C, 28.7 and 2.7 m, respectively. The visibility was approximately 2 m near the surface and 5 m around the sampling points in both piers. It took less than 10 min from sampling to Sterivex filtration.

Water samples were also collected at the same locations on December 18, 2018. Water temperature around sampling points of pier 1 and pier 2 were 16.2°C and 15.6°C, respectively. Salinity near pier 2 was 28.0, and visibility was about 1 m near the surface and 4 m around the sampling points. Except for Sterivex filtration, the process from water sampling to preservation was performed in the same manner. The process from sampling to preservation was performed within three hours.

For the Sterivex filter, we checked for a potential bias between the upper and lower layers of the Lamizip sampling bag. During the Sterivex water filtration in December, we first filtered the upper layer, and then the lower layer by using a vinyl tube. The eDNA concentrations of these two layers were compared.

## Biomass estimation based on underwater visual censuses

Underwater visual censuses by SCUBA were conducted at six locations in Otomi (Fig 1D) and six locations in Nagahama (three locations × two piers) (Fig 1C) at the time of water sampling, above. The number of individuals, body length of fishes, and umbrella diameter of jellyfish was recorded on an underwater slate in an area of approximately 100 m$^2$ (50 m by 2 m) around each water sampling point [46]. In this survey, the modified transect method termed "fin-kick transect" was applied, in which the distance traveled was estimated by the number of fin kicks made [47, 48]. Fish and jellyfish with the minimum size of 1 cm were recorded on an underwater slate in our routine survey, although the smallest individuals recorded in the present study were 3 cm. The length (L cm) of each species was converted to biomass (W g) using the length-weight relationship reported in previous studies [49–52].

$$W = aL^b$$

a: a parameter describing body shape and condition
b: a parameter for the allometric growth in body proportion

## eDNA extraction from Sterivex filters

Extraction of eDNA from Sterivex filters was performed using a DNeasy Blood & Tissue Kit (Qiagen, Hilden, Germany) based on the method of Miya et al. [31] and the Environmental DNA Sampling and Experiment Manual (Version 2.1) [45], with a slight modification (Fig 2). Each filter cartridge was centrifuged for 2 min at 4,000 × *g*. After addition of 1 mL of Bottled

Water for Molecular Biology (Merck Millipore, Darmstadt, Germany), the cartridge was centrifuged again. Then, a 220 μL solution composed of 20 μL Proteinase K and 200 μL lysis buffer (AL buffer) was added into the cartridge. The cartridge was incubated at 56˚C, 20 rpm for 20 min, and the lysed DNA solution was collected after centrifugation. After 200 μL of ethanol was added to the collected liquid, the mixture was transported to a spin column and centrifuged for 1 min at 6,000 × g. Subsequently, we followed the manufacturer's instructions and eluted in a 100 μL elution buffer (AE buffer) before preserving at -20˚C.

### eDNA extraction from GF/F

Extraction of eDNA from GF/F was performed using a DNeasy Blood & Tissue Kit according to a previous study [20] (Fig 2). Each filter was placed in a Salivette tube (Sarstedt, Nümbrecht, Germany) and centrifuged for 3 min at 5,000 × g. Then, a 420 μL solution composed of 20 μL Proteinase K, 200 μL AL buffer and 200 μL $H_2O$ was put on the filter. The tube was incubated at 56˚C for 30 min, and the lysed DNA solution was collected by centrifugation. After adding 200 μL of tris-ethylenediaminetetraacetic buffer (TE buffer) to the filter, the liquid was again collected by centrifugation. After a 200 μL AL buffer and 600 μL ethanol were added to the collected liquid, the mixture was transported to a spin column and centrifuged for 1 min at 6,000 × g. Subsequently, we followed the manufacturer's instructions and eluted in a 100 μL AE buffer before preserving at -20˚C.

### PCR analysis

Five fish species and two cnidarian species with a high frequency of occurrence in the study sites were selected for PCR analysis; blackhead seabream *Acanthopagrus schlegelii*, Japanese anchovy *Engraulis japonicus*, wrasse *Halichoeres tenuispinnis*, striped knifejaw *Oplegnathus fasciatus*, Japanese jack mackerel *Trachurus japonicus*, moon jellyfish *Aurelia aurita*, and Japanese sea nettle *Chrysaora pacifica*. The eDNA concentrations were quantified by quantitative PCR (qPCR) using a LightCycler 96 System (Roche Diagnostics, Mannheim, Germany). DNA from each target species was amplified by using the species-specific designed primers and probe sets from mitochondrial *cytochrome b* gene and partial mitochondrial *cytochrome oxidase I* gene (COI) region (Table 1). Primers/probe sets were confirmed to amplify each specific target; *E. japonicus* [53], *T. japonicus* [20 and Suppl. material 3], *C. pacifica* [18], and *A. aurita* (Yoden et al., unpublished data). The sequences of the three target species *A. schlegelii*, *H. tenuispinnis*, *O. fasciatus*, and related species known to inhabit Maizuru and Wakasa Bay [46], were collected from GenBank (see S1 Table for details of related species considered). Primer sets with more than two substitutions between the target and related species, within five bases from the 3′ end, were designed for each fish species; base pair mismatches in the 3′ end are important for primer specificity [54]. The probes were designed using Primer Express 3.0 software (Thermo Fisher Scientific, Waltham, MA, USA). The specificity of the primer sets was checked using primer-BLAST (NCBI nucleotide database) with default settings.

To confirm the specificity of the primers/probe sets, the DNA of the most closely related fish species was tested with the established real-time PCR for each species. The red seabream (*Pagrus major*), the multicolorfin rainbowfish (*Halichoeres poecilopterus*), and the rudderfish (*Girella punctata*) are the most closely related fish species to *A. schlegelii*, *H. tenuispinnis*, and *O. fasciatus*, respectively, and inhabit the same survey area [46]. Tissue samples of these fishes were mostly obtained from the Fish Collection of Kyoto University (FAKU), in which fish specimens are routinely provided from a local fish market. Two species, *G. punctata* (n = 4) and *O. fasciatus* (n = 3), were additionally collected by hook-and-line fishing or a hand net in Wakasa Bay, Sea of Japan for the present study. They were euthanized by an overdose of

**Table 1. Sequences of primers and probes for detecting eDNA of five fish and two cnidarian species targeted in this study.**

| Target species | Name of primers/probe | Sequence (5' —> 3') | Amplicon length | Reference |
|---|---|---|---|---|
| Acanthopagrus | Asc_CytB_F | CTGTCTGCCGTCCCCTACA | 129 | This study |
| schlegelii | Asc_CytB_R | TATGGCGGCTACGATAAAAGGA | | |
| | Asc_CytB_P | FAM-TCAGTTGACAACGCAACCCTAACCCG-TAMRA | | |
| Engraulis | Eja_CytB_F | GAAAAACCCACCCCCTACTCA | 115 | Ushio et al. [51] |
| japonicus | Eja_CytB_R | GTGGCCAAGCATAGTCCTAAAAG | | |
| | Eja_CytB_P | FAM-CGCAGTAGTAGACCTCCCAGCACCATCC-TAMRA | | |
| Halichoeres | Hte_CytB_F | CGCAGACGTTGTAGTCCTCACA | 113 | This study |
| tenuispinnis | Hte_CytB_R | GTGAGAAGACTAGGAATAGTATAAAGTAGATGATG | | |
| | Hte_CytB_P | FAM-CCGTACGTAATTATTGGCCAAATCGCG-TAMRA | | |
| Oplegnathus | Ofa_COI_F | GAAACTGACTCATCCCCCTCA | 166 | This study |
| fasciatus | Ofa_COI_R | CCTGCGAGAGGCGGAT | | |
| | Ofa_COI_P | FAM-TAACATGAGCTTTTGACTGCTCCCACCCTC-TAMRA | | |
| Trachurus | Tja_CytB_F | CAGATATCGCAACCGCCTTT | 127 | Yamamoto et al. [20] |
| japonicus | Tja_CytB_R | CCGATGTGAAGGTAAATGCAAA | | ; Probe was this study |
| | Tja_CytB_P | FAM-CCGTAGCACACATCTGCCGGGA-TAMRA | | |
| Aurelia aurita | Aau_COI_F | TTACTACCCCCAGCTCTGCTTT | 120 | Yoden et al., unpublished data |
| | Aau_COI_R | TACTGAACCACCGGAATGG | | |
| | Aau_COI_P | FAM-ATGAACAATTTATCCCCCCCTAAGCGCA-TAMRA | | |
| Chrysaora | Cpa_COI_F | CCCAGATATGGCTTTTCCTAGA | 231 | Minamoto et al. [18] |
| pacifica | Cpa_COI_R | TGAGTGAGCTTGTATAGCTGATA | | |
| | Cpa_COI_P | FAM-TAGGATCCTCCCTAATTG-NFQ-MGB | | |

2-phenoxy ethanol prior to dissection to obtain tissue samples. The total DNA of the related species was extracted using a DNeasy Blood & Tissue Kit according to the protocol for tissue samples, and 10 or 100 pg of the total DNA of related species was applied as a template.

Based on a previous study [20], each 20 μL reaction mixture contained 900 nM primers (forward and reverse; F/R) and 125 nM TaqMan Probe in 10 μL TaqMan Environmental Master Mix 2.0 (Thermo Fisher Scientific, Waltham, MA, USA) and 0.1 μL AmpErase Uracil N-Glycosylase (Thermo Fisher Scientific) or 10 μL FastStart Essential DNA Probes Master (Roche), and 2 μL DNA sample. Dilution series containing $3 \times 10^1$–$3 \times 10^4$ copies per PCR tube were prepared and used as quantification standards. The qPCR conditions were as follows: 2 min at 50°C, 10 min at 95°C, 55 or 60 cycles of 15 s at 95°C and 60 s at 60°C, or 10 min at 95°C, 50 cycles of 10 s at 95°C and 30 s at 60°C. Three replicates were used for each sample, and three replicate negative controls containing PCR Grade Water (Roche) instead of template DNA were included in all PCR plates. The average of the triplicates was taken to represent eDNA concentration. For each species, PCR of standard and samples obtained by GF/F and Sterivex were performed in one plate. In all the runs, $R^2$ values of calibration curves were more than 0.98, the range of slopes, Y-intercept, and PCR efficiency were between -3.94 and -3.44, 37.38 and 40.55, and 0.79 and 0.95, respectively (S2 Table). None of the PCR-negative controls or field blank controls were PCR-amplified. To reduce the risk of carry-over contamination, the pre- and post-PCR experiments were performed in independent rooms.

## Data analysis

Biomass based on visual census and eDNA concentration was $\log_{10}$ (x+1) transformed to improve homogeneity of variance. The average of three replications was used as eDNA

concentration, and all possible combinations of the sample species, sampling date and sample locations yielded 90 fish datasets and 36 jellyfish datasets. We used one observation result corresponding to each eDNA dataset for biomass data. Correlations between eDNA concentrations obtained by the two different filtration methods (GF/F and Sterivex), and between biomass based on visual census and eDNA concentration detected by GF/F and Sterivex, were analyzed by a linear mixed effect model, in which species was treated as a random effect using the "lmer" functionality of R statistical software (ver. 3.4.3) [55] for all species. Parameters were estimated by linear regression analysis (the 95% confidence and prediction limits, and analysis of intercepts and slopes of these lines) using "lm" of R for fish and jellyfish, and for each species. The choice of the most suitable model was made by using Akaike's Information Criteria (AIC) from the "ANOVA" in R. Values were removed from the linear regression analysis of biomass and eDNA concentration when either of them was 0. Differences in eDNA detected in the upper and lower layers of collection bags were evaluated using a paired Student's t-test in the R software.

## Results

### The specificity of the primers

For three targeted species, *A. schlegelii*, *H. tenuispinnis*, and *O. fasciatus*, real-time PCRs with 10 or 100 pg of the total DNA of the most related sympatric species as a template, showed no amplification in any of the three replicates. Thus, primers designed in this study had enough specificity for detecting the targeted species in our survey area.

### Comparison of eDNA concentrations obtained by GF/F and Sterivex

A linear regression equation of eDNA concentrations obtained by the two different filtration methods (GF/F and Sterivex) was calculated for all species ($Y_{Sterivex} = 0.75\ X_{GF/F} + 0.22$; Fig 3). There was a strong correlation between them in both fish and jellyfish ($R^2 = 0.74$ and $0.95$, respectively; Fig 4). The intercept of the linear regression line of fish was not significantly different from 0 ($p = 0.54$), whereas that of jellyfish was significantly higher than 0 ($p < 0.01$). The 95% confidence limits were lower than y = x when the eDNA amount was high in both taxa. The eDNA amount detected using the Sterivex filtration method was higher than that detected using the GF/F filtration method when the eDNA concentration of jellyfish was low (Fig 4). There were several cases where the eDNA concentration, in both fish and jellyfish, obtained using GF/F was 0, whereas that obtained using Sterivex was higher than 0. There were some cases where the eDNA concentration, only in fish, obtained using Sterivex was 0, but that obtained using GF/F was higher than 0. Intercepts and slopes of these lines were slightly different depending on the different fish and jellyfish taxa using "lmer" analysis (Fig 3). There was a significant difference in intercept between *A. aurita*, *C. pacifica*, *H. tenuispinnis*, *T. japonicus* and *A. schlegelii*, *E. japonicus*, *O fasciatus*. There was a difference in slopes between *A. aurita*, *C. pacifica*, *H. tenuispinnis*, *O fasciatus*, *T. japonicus* and *A. schlegelii*, *E. japonicus* using "lm" analysis ($p = 0.05$; S1 Fig). There was no significant difference in eDNA concentration between the upper and lower layer of a stable standing Lamizip ($p = 0.92$).

### Comparison between eDNA concentration and biomass

A linear regression equation of eDNA concentrations (obtained by GF/F and Sterivex), and biomass (based on underwater visual census) was calculated for all species ($Y_{eDNAconc} = 0.58\ X_W + 1.52$; Fig 5). There was a strong positive correlation between eDNA and fish biomass ($R^2 = 0.64$; Fig 6A). A similar but weaker correlation was observed between eDNA and jellyfish

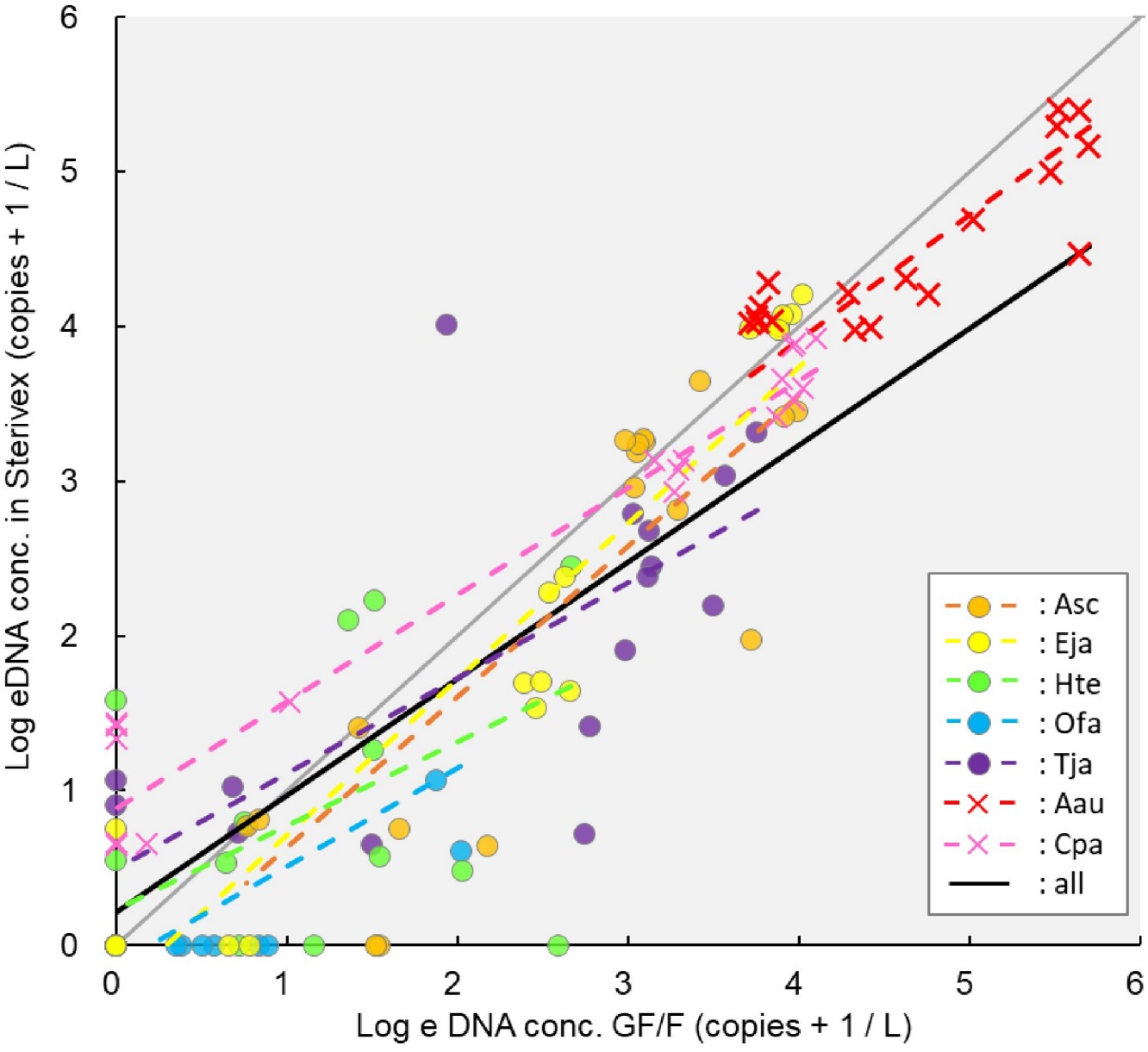

**Fig 3. Correlation of eDNA concentration estimated by two different filtration methods (GF/F and Sterivex) in all species.** Linear regression equation including all species: $Y_{Sterivex} = 0.75\ X_{GF/F} + 0.22$, gray lines: y = x, Asc: *Acanthopagrus schlegelii*, Eja: *Engraulis japonicus*, Hte: *Halichoeres tenuispinnis*, Ofa: *Oplegnathus fasciatus*, Tja: *Trachurus japonicus*, Aau: *Aurelia aurita*, Cpa: *Chrysaora pacifica*.

($R^2 = 0.27$; Fig 6C). There was no significant difference between the slopes of these two lines ($p = 0.26$), whereas the intercepts were significantly higher than 0 ($p < 0.01$), and those of jellyfish were significantly higher than those of fish ($p < 0.01$). The $Log_{10}$ (x+1) transformed copy numbers of eDNA concentration per biomass in jellyfish (intercept: 3.55) were much greater than those in fish (intercept: 0.19), representing an eDNA emission rate which was about 700 times (344–991 times) higher in jellyfish than in fish. The correlation between biomass and eDNA concentration showed no difference between GF/F (S4 Fig) and Sterivex (S2 Fig) either in fish ($R^2 = 0.64$; Fig 6A and $R^2 = 0.60$; Fig 6B) or in jellyfish ($R^2 = 0.27$; Fig 6C and $R^2 = 0.24$; Fig 6D). Intercepts of regression lines were different depending on species.

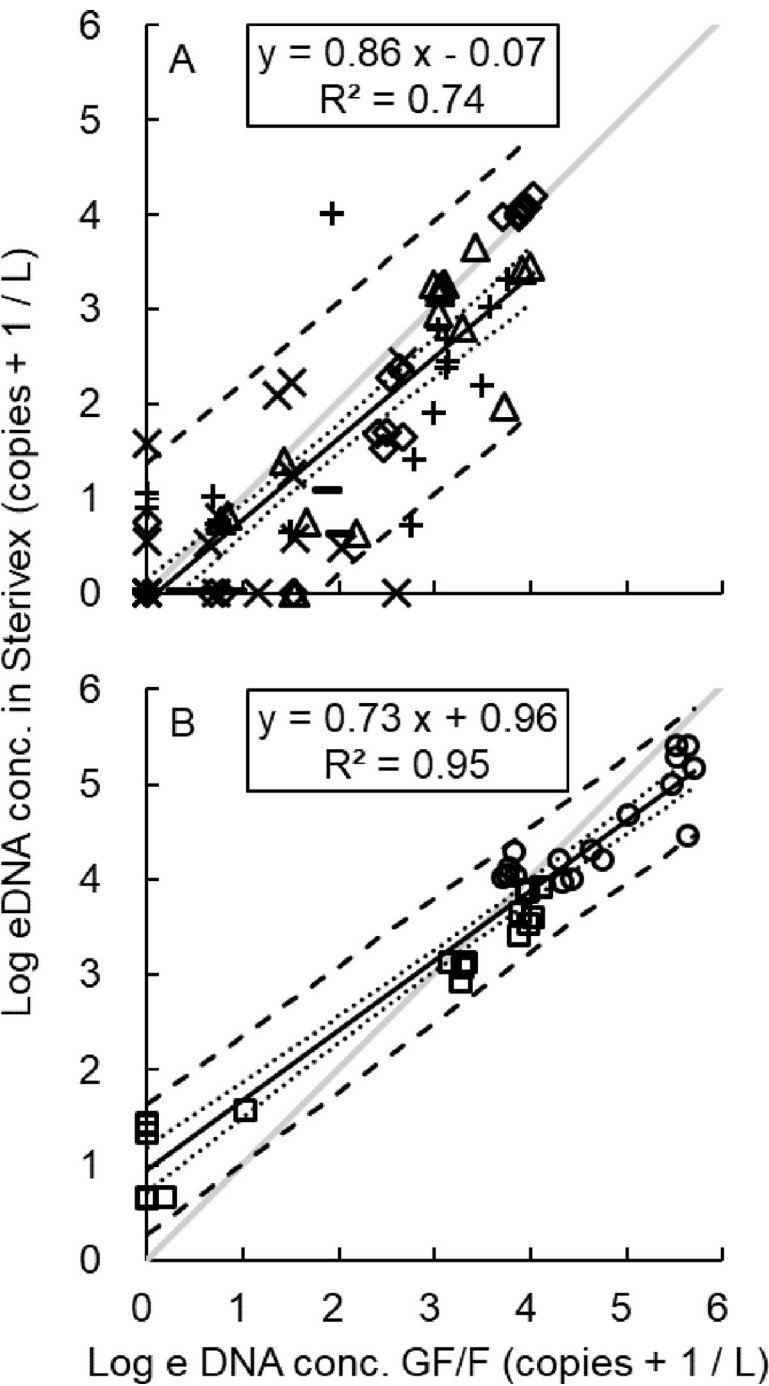

**Fig 4.** Correlation of eDNA concentration estimated by two different filtration methods (GF/F and Sterivex) in fish (A) and jellyfish (B). (A) fish, △: *Acanthopagrus schlegelii*, ◇: *Engraulis japonicus*, ×: *Halichoeres tenuispinnis*, −: *Oplegnathus fasciatus*, +: *Trachurus japonicus*, (B) jellyfish, ○: *Aurelia aurita*, □: *Chrysaora pacifica*, dotted lines: the 95% confidence limits, dashed lines: the 95% prediction limits, gray lines: y = x.

Out of all the 126 sampling stations, 33 (26%) were positive in both the eDNA analysis and the visual census, 26 (21%) were negative in both, 61 (48%) were positive only in the eDNA analysis, and 6 (5%) were positive only in the visual census. Each species was detected by

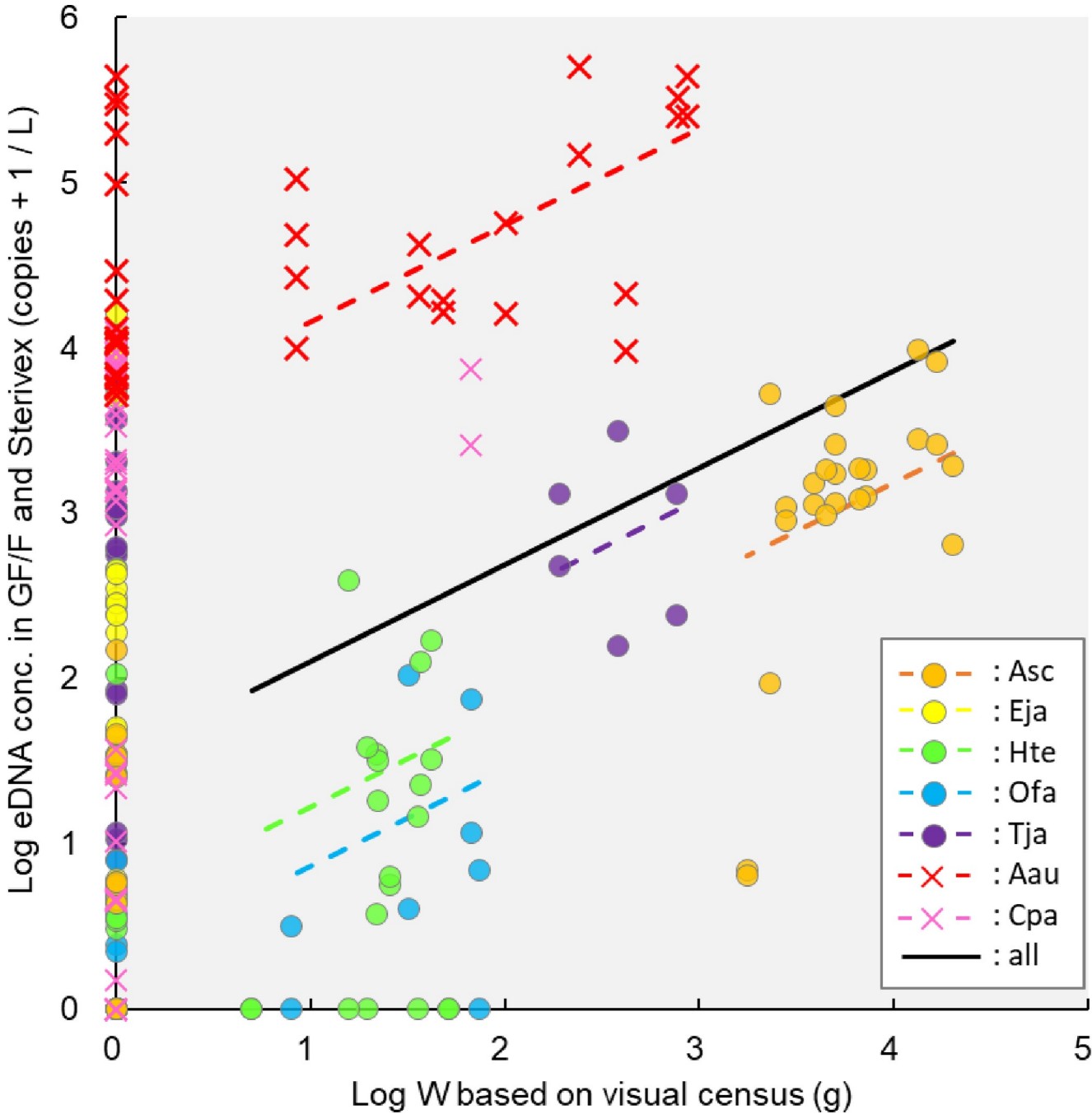

**Fig 5. Correlation between biomass (W) and eDNA concentration estimated by GF/F and Sterivex.** Linear regression equation including all species: $Y_{eDNAconc} = 0.58 X_W + 1.52$, Asc: *Acanthopagrus schlegelii*, Eja: *Engraulis japonicus*, Hte: *Halichoeres tenuispinnis*, Ofa: *Oplegnathus fasciatus*, Tja: *Trachurus japonicus*, Aau: *Aurelia aurita*, Cpa: *Chrysaora pacifica*.

eDNA analysis but not by visual census at least once, whereas the opposite was true only in *H. tenuispinnis* and *O. fasciatus* (Fig 6).

A large amount of *C. pacifica* eDNA was detected at the point where one individual of this species was observed underwater, and smaller amounts were detected at other points (S3 Table; S3 and S5 Figs). *Aurelia aurita* eDNA was abundant at every point (S3 and S5 Figs).

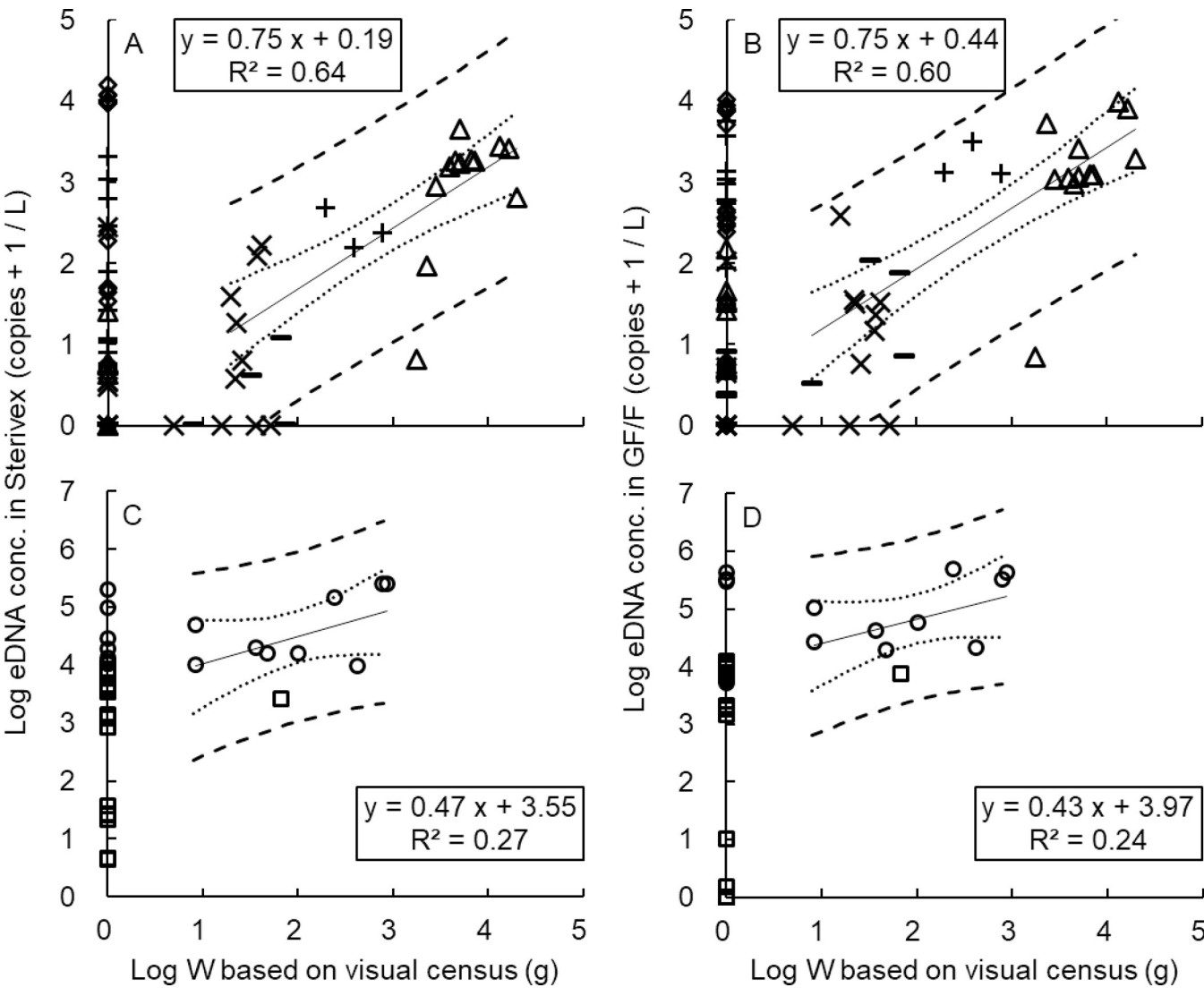

**Fig 6.** Correlation between biomass (W) and eDNA concentration estimated by Sterivex (A, C) and GF/F (B, D) in fish (A, B) and jellyfish (C, D). (A, B) fish, △: *Acanthopagrus schlegelii*, ◇: *Engraulis japonicus*, ✕: *Halichoeres tenuispinnis*, −: *Oplegnathus fasciatus*, +: *Trachurus japonicus*, (C, D) jellyfish, ○: *Aurelia aurita*, □: *Chrysaora pacifica*; dotted lines: the 95% confidence limits, dashed lines: the 95% prediction limits.

Although live individuals were sparse (S3 Table; S3 and S5 Figs), many dead individuals were found on the seafloor, especially in Otomi in May (Fig 1B). Amounts of eDNA in *A. schlegelii*, *H. tenuispinnis*, *O. fasciatus* and *T. japonicus* were larger at the points where biomass was greater, and smaller where biomass was less, in Nagahama in September (S3 Table; S3 and S5 Figs).

Detection of eDNA and visually detected abundance in jellyfish, *H. tenuispinnis*, *O. fasciatus* and *T. japonicus* was larger in September than in December (S3 Table; S3 and S5 Figs). Between the two species of jellyfish, the eDNA of *A. aurita* was consistently larger than that of *C. pacifica* (S3 and S5 Figs). Detection rates of eDNA in *A. schlegelii*, *E. japonicus* and *T. japonicus* were higher than those in *H. tenuispinnis* and *O. fasciatus* (S3 Table; S3 and S5 Figs).

## Discussion

### Comparison of performance between GF/F and Sterivex

There was a clear correlation between the eDNA concentrations obtained by GF/F and Sterivex, and the linear regression equation including all the species was $Y_{Sterivex} = 0.75\ X_{GFF} + 0.22$ (Fig 3). In the present study, the amount of eDNA detected using the Sterivex filtration method was higher than that detected using the GF/F filtration method when the eDNA concentration of jellyfish was relatively low (Fig 4B). These results suggest that Sterivex may have superior sensitivity for eDNA. In previous studies on freshwater fish, Sterivex yielded higher amounts of eDNA, represented by lower cycle quantification (Cq) values, than did GF/F [39]. In addition, for eDNA metabarcoding, the extracted amount of eDNA was lower and the detection rate of fish species was higher using Sterivex than by using open filtration methods [38]; the number of species detected by Sterivex was approximately 1.5 times higher than those detected by GF/F [31]. On the other hand, the amount of eDNA detected by GF/F was higher than that detected by Sterivex when the eDNA concentration was higher, both in fish and jellyfish (Figs 3 and 4). These results suggest that GF/F may be superior when large amounts of eDNA need to be extracted. As the pore size of Sterivex (0.45 μm-pore size) was smaller than that of the GF/F (0.7 μm-pore size), the eDNA amount captured by Sterivex was expected to be greater than that captured by the GF/F; nevertheless, this was not the case in this study. There may be four possible explanations for this: i) the difference of eDNA concentration between the upper and lower layers in water sampling bags, ii) filtration processes, iii) extraction losses, and iv) species differences.

Regarding explanation i), it was reported that eDNA concentrations of fish [14] and jellyfish [18] were higher near the bottom than near the surface; another study [56] reports that the detection rate of eDNA was not significantly different between the near-surface and subsurface. Urine, slimy coatings, saliva, dead carcasses, and predator and prey feces [2, 25] have all been suggested as possible eDNA origins; the eDNA state in the water may be free, cellular or particle-bound [9]. The amount of eDNA in the lower layer is suspected to be higher than that in the upper layer because larger particles sink faster than smaller ones [57]. However, in the present study there was no significant difference in eDNA concentrations between the two layers in the sampling bags ($p = 0.92$). In the method of this study, Sterivex samples were filtered from the upper layer and GF/F from lower layer of water sampling bags. The difference between the layers is not likely to have caused any systematic difference between the two filtration methods. Thus, the difference of detection tendency between the two methods should be attributed to filtration and/or extraction processes.

Regarding explanation ii), it is possible that the amount of eDNA detected depends on how one applies pressure during the filtration process. There are many variables one needs to consider when applying a filtration method [25, 26]. The choice of the filter paper types can substantially affect DNA yield, depending on eDNA binding capacity [30]. Finer filters tend to clog, and therefore either require a longer time, or are unable to filter a sufficient volume of water [28, 38, 40]. Too much filtration power in relation to filter hole size causes samples to leak from the filter funnels during filtration [30]. In this study, GF/F filtration was conducted with negative pressure using an aspirator, while Sterivex filtration was conducted with positive pressure using a syringe. The difference between negative and positive pressure may cause differences in the amount of detected eDNA.

Regarding explanation iii), extraction losses may offer a possible explanation as to why the eDNA concentrations obtained by the two filtration methods were different. It is known that the size distribution of eDNA particles varies with environmental conditions [40]. The amount of detected eDNA depends on different eDNA characteristics (size, spatial structure, extra-

and intracellular, and particle-bound and free) [30], indicating that the optimal filter choice varies for different extraction methods and there are different protocol combinations suitable for different organisms [28, 58]. At the time of extraction, it is necessary for DNA to flow out of cells, and be removed from filter paper by elution and centrifugation, so that it does not remain on the walls of the tube and column, resulting in eDNA loss. Furthermore, the procedure of eDNA extraction using the GF/F and Sterivex filtration method was similar, but not exactly the same, during the experiments presented here, which was a necessity to compare the two methods (Fig 2). Therefore, it is possible that these differences in methods may contribute to differences detected in the results. Specifically, BAC was used for DNA preservation of the GF/F samples, and RNA*later* was used for DNA preservation of the Sterivex filtrations. BAC is cationic surfactant and reduces microbial activity [59], whereas RNA*later* is a stable reagent which deactivates nucleases [60]. Water samples with BAC are reported to retain more than 92% of fish DNA for 8-h at ambient temperature [59], and those with RNA*later* are reported to successfully preserve the same amount of planktonic DNA for over 1 month at ambient temperature as frozen samples [60]. This study conducted the process from sampling to preservation in a freezer in less than seven hours, and the samples were transported on ice immediately after the addition of BAC or RNA*later*. Therefore, we assume that there was not much difference in the result due to the preservation step. However, it is also reported that fish species detected by metabarcoding analysis using BAC were lower than those stored on ice [61], whereas RNA*later* yielded a substantial precipitate that inhibited qPCR amplification of fish [62]. RNA*later* has been shown to store good quality DNA, but not always in high enough quantities for metabarcoding analysis of marine organisms [63]. Therefore, the difference between BAC and RNA*later* preservation may be a subject that requires further investigation.

Regarding explanation iv), species differences, intercepts and slopes of linear regression lines, representing the correlation of eDNA concentrations to the two filtration methods, were slightly different depending on fish and jellyfish species (S1 Fig). A few fish species had different parameters for intercepts and slopes compared to the other fish species, but remained closer to the parameters of jellyfish (Fig 3). Such differences may be related to the "ecology" of eDNA, i.e., myriad interactions between extraorganismal genetic material and its environment [2], which would be variable among species.

## Accuracy of estimating biomass of marine species by eDNA

A positive correlation between eDNA concentration and biomass has been reported in freshwater amphibians [9, 10] and fish [11, 12, 19, 21], and marine fish [4, 6, 20] and jellyfish [18]. It is possible to quantify temporal variation, seasonal changes and annual fluctuation of marine species based on underwater visual census [46, 48, 64]. In the present study, there was also a positive correlation between eDNA concentration and biomass estimation based on visual observation in all species ($Y_{eDNAconc} = 0.58 X_W + 1.52$; Fig 5), and in both fish and jellyfish ($R^2 = 0.64$, $R^2 = 0.27$; Fig 6A and 6C). Furthermore, eDNA concentration per biomass in jellyfish was approximately 700 times greater than it was in fish. Our results indicate that the detection or release rate of eDNA may be different depending on the target species.

Focusing on the differences in the detection or release rate of eDNA for each target species, the following factors can be considered. Minamoto et al. [18] reported that the eDNA concentration of *C. pacifica* was ~13 times higher near the bottom than on the surface. In the present study, the eDNA concentration was higher at the point where *C. pacifica* was visually detected, and the eDNA of this species was also detected at other points (S3 Table; S2 Fig). We collected samples near the bottom; therefore, eDNA released from individuals who passed through before our visual survey was likely to be detected. The long tentacles of this species are easily

torn and can, therefore, be a major source of eDNA. For *A. aurita*, the eDNA concentration was high at every point (S2 Fig). Especially in Otomi in May, the eDNA was positive at the points where live individuals were not observed (S3 Table; S3 and S5 Figs) whereas many dead individuals were found on the bottom (Fig 1B). Since *A. aurita* is often eaten by fishes [65], many dead and torn individuals drift in the sea. Merkes et al. [66] reported that eDNA released from dead fish was detected for more than one month. Thus, eDNA may over-estimate the abundance of species when high mortality occurs nearby. Furthermore, various life stage of jellyfish other than medusae, i.e., eggs, planulae and polyps, can be a source of eDNA but would have been missed in visual census. Generally, a high correlation between eDNA concentration and biomass exists in fishes (Fig 6A). Larger amounts of eDNA in *A. schlegelii*, *H. tenuispinnis*, *O. fasciatus*, and *T. japonicus* were detected at points where visually evaluated biomass was greater (S3 Table; S2 Fig). This may be related to lifestyle issues discussed below.

It has been reported that the presence/absence of aquatic species can be monitored using eDNA analysis, even for non-native species [3], threatened species [37, 41], and fish in mountainous rivers during the winter [42]. It is also known that eDNA detection does not necessarily correlate with the presence of organisms [22]. This was also the case in 53% of the samples examined in the present study. Potential causes of this could be currents, seasonality, and animal activity, including their life stage.

A positive correlation between biomass and eDNA concentration has been reported in rivers [10, 21] and the sea when there are currents [6, 18–20], although it has been pointed out that currents may influence eDNA quantification [4, 6]. When examining the eDNA at a certain location and time, the outcome may depend on whether the source is up or down the current. For instance, in a river, eDNA has been detected in a range of 240 m to 12 km downstream [43, 67–69]. Sansom and Sassoubre [69] reported that, in theory, eDNA could be transported for 4.3–36.7 km downstream. In the sea, eDNA has been detected in a range of 10 m to 150 m from its source [20, 70, 71]. It is likely that eDNA, released from dead or injured *A. aurita* and *C. pacifica* in adjacent waters, or from *T. japonicus* schooling and migrating just before observation (e.g., Fig 2), or from large amounts of small size individuals (such as gametes or larvae), drifted in the range and was collected. In one instance, *H. tenuispinnis* and *O. fasciatus* were detected by visual census but not by eDNA analysis (Fig 6A). Both species are demersal (Fig 2), and water was sampled at 1 m above the sea bottom. A relatively small size and abundance, as well as complex water movement, may have hindered eDNA detection in this case.

Seasonal fluctuations of eDNA amounts have been reported, and they are consistent with seasonal variations of biomass in freshwater fishes [14, 19, 21] and marine jellyfish [18]. The seasonal change of fish species in an estuary can also be detected by eDNA metabarcoding analysis [72]. In the present study, detection of eDNA and individuals of *H. tenuispinnis*, *O. fasciatus*, *T. japonicus*, and *C. pacifica*, which occur in the surveyed area from spring to summer, was larger in September than in December (S3 Table; S4 and S5 Figs). A dense patch of *A. aurita* is often found north, off the coast of the survey point in Nagahama, and it occasionally comes close to the shore (Masuda R, *pers. obs.*). This suggests that biomass quantification of *A. aurita* using eDNA may be more feasible in a larger spatial scale (such as off coast where the area may be less affected by many dead individuals or non-medusa stages, as would happen closer to the coast) and/or with temporal change. eDNA is more likely to capture such a seasonal fluctuation of biomass than the spatial variation.

The activity variation of organisms may also influence the seasonal fluctuation of eDNA. eDNA emission increases by feeding [12, 73] and reproduction [24, 74, 75], and is dependent on the lifestyle of the target species [69]. In the present study, the detection rate of eDNA was different between fish species having different lifestyles (e.g., Fig 2). The detection rate of

eDNA may be high in *E. japonicus* and *T. japonicus*, of which many, small sized individuals form schools of several hundreds and migrate near the survey point. The eDNA detection rate may also be high in *A. schlegelii*, of which a small number of large sized individuals are consistently found in the shallow reef area. eDNA detection rates may be low in *H. tenuispinnis* and *O. fasciatus*, as they are small in size, few in numbers and demersal (S3 Table; S2 Fig). There is a possibility that the difference in activity of each species and difference in behavioral traits (such as bold-shy behavior [76]), could affect eDNA release, and can be a subject of future study.

Furthermore, because eDNA could not be detected when the population density is very low [3] and eDNA concentration is very low in seawater samples, it may be predicted that some samples will be negative in eDNA. Possible factors of mismatch between detection in observation and non-detection in eDNA are PCR amplification inhibition, interference of non-target species [25, 77], and mixing of different haplotypes in the same waters (Takahashi et al., unpublished data). Unlike in aquaria or ponds, eDNA may not be detected in marine environments even if the target species is found in them.

eDNA detection does not necessarily correlate with the presence of certain organisms in complex oceanic environmental conditions. Therefore, when it comes to estimating biomass, it is better to consider data obtained from several samplings, than to just rely on visual observation at the time of each water sampling. At the time of the underwater visual census in Otomi, there was a point where some *H. tenuispinnis* individuals were found in the return path and not in the forward path where water was sampled (Fig 1A and 1D; S3 Table). In Nagahama in December, one *A. aurita* with a 10-cm umbrella diameter was observed near the pier 1 at about 1 h before the census of this study (Fig 1A and 1C). By combining several observations, it is expected that the correlation between biomass and eDNA amount will be improved. Furthermore, by knowing the behavior patterns and physiology of target species as well as the characteristics of the habitat, it is possible to estimate the biomass of marine organisms with a higher accuracy when performing eDNA analysis by the GF/F or Sterivex filtration methods.

## Conclusion

This study evaluated the eDNA concentration obtained by two different filtration methods (GF/F and Sterivex). A comparison with an underwater visual census also showed a positive relationship between the eDNA concentration and the fish and jellyfish biomass. It is possible to convert data obtained by GF/F to that obtained by Sterivex. We found that some species are detected more easily by eDNA, while a small number of other species showed an opposite trend. This was likely to be due to the size of the target species and their lifestyle (such as pelagic or demersal), seasonality and behavior, as well as physical factors such as water currents. Therefore, we suggest that the eDNA method can be particularly effective in combination with knowledge of the ecology, behavior and life history of the target species. Such an approach is expected to give us further ecological insights that would be valuable for the conservation of the ocean environment and the management of fisheries resources [72, 78].

## Supporting information

**S1 Table. List of related species used in the *in silico* specificity test and further checking with real-time PCR.** The most closely related species within the same order were checked since no fish species belonging to the same family are present in the surveyed area. *Asc: *Acanthopagrus schlegelii*, Hte: *Halichoeres tenuispinnis*, Ofa: *Oplegnathus fasciatus*. ** X indicates specificity was checked.
(XLSX)

**S2 Table. Parameters of the standard curve for each species at each sampling site and date.**
*Asc: *Acanthopagrus schlegelii*, Eja: *Engraulis japonicus*, Hte: *Halichoeres tenuispinnis*, Ofa: *Oplegnathus fasciatus*, Tja: *Trachurus japonicus*, Aau: *Aurelia aurita*, Cpa: *Chrysaora pacifica*.
(XLSX)

**S3 Table. Body length (L cm) and estimated biomass (W g) of target organisms encountered in underwater observation.** *Asc: *Acanthopagrus schlegelii*, Eja: *Engraulis japonicus*, Hte: *Halichoeres tenuispinnis*, Ofa: *Oplegnathus fasciatus*, Tja: *Trachurus japonicus*, Aau: *Aurelia aurita*, Cpa: *Chrysaora pacifica*. **$W = aL^b$ was used as weight-length relationships based on the FishBase website [49]. The values of Ofa were based on those of the same genus, *O. woodwardi*, and Aau based on Aoki et al. [51] and Cpa based on Yasuda [50]. ***L: length, N: number, W: weight. ****Points and lines were shown in Fig 1. *****Tentacles of Cpa were torn.
(XLSX)

**S4 Table. Raw data used in Figs 3, 4, 5 and 6 and S1 and S2 Figs.**
(XLSX)

**S1 Fig. Correlation of eDNA concentration between GF/F and Sterivex for each species in water samples collected on May 15 in Otomi, on September 19 and December 18 in Nagahama.** Asc: *Acanthopagrus schlegelii*, Eja: *Engraulis japonicus*, Hte: *Halichoeres tenuispinnis*, Ofa: *Oplegnathus fasciatus*, Tja: *Trachurus japonicus*, Aau: *Aurelia aurita*, Cpa: *Chrysaora pacifica*, gray lines: y = x.
(TIF)

**S2 Fig. Correlation between biomass (W) and eDNA concentration estimated by Sterivex.** Linear regression equation including all species: $Y_{eDNAconc} = 0.63 X_W + 1.21$, Asc: *Acanthopagrus schlegelii*, Eja: *Engraulis japonicus*, Hte: *Halichoeres tenuispinnis*, Ofa: *Oplegnathus fasciatus*, Tja: *Trachurus japonicus*, Aau: *Aurelia aurita*, Cpa: *Chrysaora pacifica*.
(TIF)

**S3 Fig. Correlation between estimated biomass (W) and eDNA concentration obtained by Sterivex for each species in water samples and biomass data collected on May 15 in Otomi, on September 19 and December 18 in Nagahama.** Abbreviations are the same as in S2 Fig.
(TIF)

**S4 Fig. Correlation between biomass (W) and eDNA concentration estimated by GF/F.** Linear regression equation including all species: $Y_{eDNAconc} = 0.55 X_W + 1.77$, abbreviations are the same as in S2 Fig.
(TIF)

**S5 Fig. Correlation between estimated biomass (W) and eDNA concentration obtained by GF/F for each species in water samples and biomass data collected on May 15 in Otomi, on September 19 and December 18 in Nagahama.** Abbreviations are the same as in S2 Fig.
(TIF)

## Acknowledgments

We would like to thank Dr. Hiroki Yamanaka (Ryukoku University) for his advice on the eDNA extraction technique from Sterivex and Mr. Takaya Yoden, Mr. Tatsuki Toya, and Ms. Misaki Shiomi (Kyoto University) for their help in water sampling and filtration. The fish tissue samples archived in the Fish Collection of Kyoto University (FAKU) were kindly provided

by Drs. Yoshiaki Kai and Fumihito Tashiro at Maizuru Fisheries Research Station, Kyoto University. We would also like to thank Dr. Ruslan Kalendar (University of Helsinki) and three anonymous reviewers for their constructive comments that helped us to substantially improve the quality of the manuscript.

## Author Contributions

**Conceptualization:** Sayaka Takahashi.

**Data curation:** Sayaka Takahashi, Reiji Masuda.

**Formal analysis:** Sayaka Takahashi.

**Funding acquisition:** Reiji Masuda.

**Investigation:** Sayaka Takahashi, Reiji Masuda.

**Methodology:** Sayaka Takahashi, Masayuki K. Sakata, Toshifumi Minamoto, Reiji Masuda.

**Project administration:** Sayaka Takahashi, Reiji Masuda.

**Resources:** Reiji Masuda.

**Validation:** Sayaka Takahashi, Masayuki K. Sakata, Toshifumi Minamoto.

**Visualization:** Sayaka Takahashi.

**Writing – original draft:** Sayaka Takahashi.

**Writing – review & editing:** Sayaka Takahashi, Masayuki K. Sakata, Toshifumi Minamoto, Reiji Masuda.

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
