## [Decision Letter · Decision Letter 0]

31 Dec 2019

PONE-D-19-32313

Comparing the efficiency of open and enclosed filtering systems in fish and jellyfish environmental DNA quantification

PLOS ONE

Dear Dr. Takahashi,

Thank you for submitting your manuscript to PLOS ONE. After careful consideration, we feel that it has merit but does not fully meet PLOS ONE’s publication criteria as it currently stands. Therefore, we invite you to submit a revised version of the manuscript that addresses the points raised during the review process.

We would appreciate receiving your revised manuscript by Feb 14 2020 11:59PM. To enhance the reproducibility of your results, we recommend that if applicable you deposit your laboratory protocols in protocols.io, where a protocol can be assigned its own identifier (DOI) such that it can be cited independently in the future. For instructions see: http://journals.plos.org/plosone/s/submission-guidelines#loc-laboratory-protocols

We look forward to receiving your revised manuscript.

Kind regards,

Ruslan Kalendar, PhD

Academic Editor

PLOS ONE

Journal Requirements:

2. We note that you are reporting an analysis of a microarray, next-generation sequencing, or deep sequencing data set. PLOS requires that authors comply with field-specific standards for preparation, recording, and deposition of data in repositories appropriate to their field. Please upload these data to a stable, public repository (such as ArrayExpress, Gene Expression Omnibus (GEO), DNA Data Bank of Japan (DDBJ), NCBI GenBank, NCBI Sequence Read Archive, or EMBL Nucleotide Sequence Database (ENA)). In your revised cover letter, please provide the relevant accession numbers that may be used to access these data. For a full list of recommended repositories, see http://journals.plos.org/plosone/s/data-availability#loc-omics or http://journals.plos.org/plosone/s/data-availability#loc-sequencing.

Reviewers' comments:

Reviewer's Responses to Questions

**Comments to the Author**

1. Is the manuscript technically sound, and do the data support the conclusions?

Reviewer #1: No

Reviewer #2: Yes

Reviewer #3: Yes

2. Has the statistical analysis been performed appropriately and rigorously? 

Reviewer #1: No

Reviewer #2: Yes

Reviewer #3: Yes

3. Have the authors made all data underlying the findings in their manuscript fully available?

Reviewer #1: Yes

Reviewer #2: Yes

Reviewer #3: Yes

4. Is the manuscript presented in an intelligible fashion and written in standard English?

Reviewer #1: Yes

Reviewer #2: Yes

Reviewer #3: Yes

5. Review Comments to the Author

Reviewer #1:

Takahashi and colleagues present the results of an experiment comparing the effects of two different extraction methods — Sterivex and Glass-Fiber Filters — on an eDNA study of fish and jellyfish from the field. In particular, they evaluate the suitability of these two filtering methods in studies comparing species-specific qPCR results to visual counts of biomass. Reassuringly, they show the two filtering methods generally agree with respect to species-specific DNA concentration, and they include some speculation about the benefits of each filtering method. This kind of study is important as eDNA methods work themselves out, and the work is generally applicable beyond the specific case of fish/jellyfish assays. However, my main concerns are statistical: given the data in hand, the authors need to ensure they have a good statistical grounding for their analyses, and that their results are presented in a rigorous way. By pooling across species and engaging in simple linear regression when the data seem to call for some kind of hierarchical model, the authors have smoothed over important difficulties in their dataset.

Line 77-79: distinguish between open and closed filters; define terms

Line 88 and elsewhere: lower-case Greek letter (phi); perhaps accidental?

Lines 134-136: Might the treatment with BAC vs. RNA Later make any difference in the results?

Lines 292-293: Here begin the statistical issues I have questions about. The analysis and results shown in Figure 3 have several problems, I think:

1. The authors are pooling across species in their linear regression. This alone would seem to violate the assumption of independence among data points, but in addition, they have pooled selectively: even if pooling were acceptable, what would be the justification for splitting apart fish and jellyfish? (Individual species are shown in the supplement, but no statistics are shown, and the un-pooled results are not nearly as strong as the pooled results suggest).

2. The hypothesis being tested here is that, for a given species, the concentration of eDNA detected from samples with GFF and Sterivex are correlated. Accordingly, each species must be treated separately (which will then show, for example, that Halichoeres seems to behave quite differently than the other species tested, etc). Or, if not separately, the authors could do a mixed-effects model in which species is treated as a random effect.

3. It is also not clear that the assumption of normality is met, even after log-transformation, but this is probably a much more minor issue.

Lines 311-315: if the authors are testing the hypothesis that the eDNA concentrations of the upper and lower parts of the same bag are different, they should do that explicitly. Moreover, by my math, there is a HIGHLY significant correlation between upper and lower samples (treating each species as a single data point of (x,y) with x being upper concentration and y being lower concentration, the slope is 0.98 and r-squared 0.98, or a nearly perfect relationship). And in any case, later in the manuscript the authors determine upper and lower not to matter at all, so why report these results?

Lines 321 - 332 and Figure 4: Same major pooling problem as before, plus:

1. Does the regression not count the zero values? That is what the lines in Figure 4 imply.

2. The plots indicate substantial Sterivex eDNA with no accompanying visual counts. This may be due EITHER to a) much visually-unobserved biomass, OR b) a high false-positive rate. Given the qPCR assay tests, what is the observed false-positive rate? And then, the authors should discuss the observation in this light.

The equivalent GFF result figure is in the supplement (S2, disaggregated into species, as it should be), but this is a core result of the paper, so it should be in the main piece.

Lines 368 - 370: the authors show no data on *total* eDNA collected, so this hypothesis is untestable. The raw qbit (or similar) results would allow them to test the idea that a larger total amount of eDNA results from a smaller pore size.

Overall, why not model the concentration of eDNA as a function of biomass (fixed effect) given species and extraction method (random effects)? This would combine all data into a single, clear and more powerful analysis.

Reviewer #2:

The authors compared two extraction methods for eDNA for jellyfish and fish in two sites. The manuscript is well written.

The experimental design and data analysis were both solid.

Discussion is detailed and very organized.

The conclusion is generally solid. Overall this is a well-conducted study.

There are several areas of improvement for the manuscript.

Four possible explanations on method differences were provided. One of the main differences if the addition of BAC in GF/F. The authors relied on one reference in the literature, which clearly showed that BAC significantly affected eDNA extraction efficiency. Therefore it is likely that the observed result difference was from BAC. However, no control experiments were conducted to compare the effects of the addition of BAC in Strerivex or effects of different concentration of BAC on GF/F. At the minimum, the authors should discuss the mechanisms of BAC of DNA preservation on effects on DNA extraction efficiency.

Another possibility on eDNA extraction efficiency is the difference eDNA extraction procedure as shown in Figure 2. They are similar but not the same ans therefore could still contribute to different results. This should be discussed as well.

Please provide melting curve and amplification efficiency of PCR analyses.

Reviewer #3:

This manuscript quantifies the efficiency and capacity of two eDNA analysis techniques and compares them to existing observational biomass census techniques. Their results indicate that there is a positive correlation between the two tested eDNA sampling methods, GF/F and Sterivex, and these two methods were also positively correlated with an underwater visual census. This information is extremely valuable in this new era eDNA research and method development. Overall I found this manuscript very well written and well presented, and I believe this manuscript has strong merit for publication with some reasonable minor changes/edits and some inclusions of finer detail in places. I have provided specific comments and suggestions below, however, please do not feel discouraged by the number of them or how they may be worded, this is in my opinion a very good manuscript.

The number refers to the line number in the original manuscript, followed by the comment.

18 Strange font

41-42 “more simple” reword. Possibly “more efficient” or similar

42-43 “After Ficetola [3] applied this method to detect bullfrogs in ponds” consider rewording so that ‘this’ is not the subject of the sentence. “After Ficetola [3] applied a newly developed eDNA method to detect bullfrogs in ponds” or similar.

52 “and tanks [11-12]. eDNA” usually try not to use abbreviations at the beginning of a sentence

57 “visual observation on shore or board [18],” reword this, suggest “visual observations detected via land or vessel based surveys”

60 “and mark-recapture techniques” not quite the right word, try “and during mark-recapture experiments”

67 “species, and this is problematic.” Change to “species, which is problematic.”

67-70 Reword this sentence so that it does not include “we”. Typically write in the third person and not with “we” as the subject of the sentence. Try something like “These studies elucidate the necessity to choose suitable filters”

86 Similar comment as above, instead of “and we also used” try “which were also used”

86 Where possible, try to avoid starting a sentence with an abbreviation (eDNA)

90-92 It is uncommon to begin a paragraph with a question. Consider changing to “To determine if there are any differences” and edit the sentence to suit.

94 change to third person rather than use “us” as the subject. Try “will facilitate the ability to estimate” or similar.

96-100 There was the additional aim whereby the upper layer and the lower layer of a sample was compared. I suggest adding it into the aims.

109 a space is required at the end of the sentence

121 change “Seawater samples, of volume 3 L each” to “Seawater samples (3 L each)”

121 please explain what a Lamizip is on the first mention, also, is this an adaptation of an existing method of water sample collection? If so please site it.

124 how were these water quality quantified. For example, was the salinity measured with a probe or refractometer, also, it is more common to report salinity in parts per thousand (ppt or ‰), practical salinity units (psu). Also, how was visibility measured; visually estimated, or with a secchi disk?

127 This figure and/or figure legend requires more detail and I am not sure how relevant the picture of dead jellyfish is to the sample site. If it is retained, explain the importance of the black arrow, add a scale bar, and label the jellyfish as most people will not know what they are looking at. Please label each of the sample sites to match the text and supplementary material (Otomi and Nagahama), explain what P1 to P6 are, explain the grey curved arrow, describe what L1-L12 are. Each figure should be able to stand alone, I mean this in the nicest way. These suggestions apply to line 127 and to figure 1.

130 This figure legend could also use a bit more detail. Consider adding text to clarify the abbreviations, and consider briefly describing each step in the legend. Each figure should be able to stand alone. I am also not sure what the top half of the diagram is there for either, other than looking nice, there is no other information than a diver conducted a visual survey during sample collection.

133-137 I would imagine that many of these steps are part of a previously published method. Please state this and site the relevant sources. This will give the methods more credibility. If these are new methods, please explain the importance of each step and/or why it was included.

155 “salinity in those piers” change to “salinity near each pier”

156-157 explain which value was associated with which pier (1or2). Explain how the values were measured (as previously suggested) (see comment line 124)

158 change to “in the same manner as the Otomi site presented above.”

161 similar comment for salinity units, and which pier was associated with these measurements, as above there were individual values for each pier.

171-177 I think the limits of the visual censuses should be included so that the minimum animal size that was detected. For example, no fish were detected under 40 mm in length or jellyfish under 50 mm, even though small specimens may be present.

205-206 Try to avoid “The procedure that followed was the same as the above.” and briefly explain the steps as there are a lot of procedures above.

218 “as being able to amplify specifically each target” suggest reword to amplify each specific target”

235 “provided from local fish market.” Reword “provided from a local fish market.”

243 consider including more information in the table legend that briefly describes the contents of the table

252-262 are any of these existing and/or published methods? If so, I suggest the relevant sources are cited which will also add credibility to performing the PCR analysis.

273 Excel is not considered the best program for statistical analysis and it may be worth checking the results in another program, such as R.

275 consider rewording the sentence so that it does not start with “S1”

285 consider writing in the third person and avoid “our primers” as the sentence subject.

295 reword to “higher than 0”

314 please add labels to these values similar to what is above on line 312-313

325 “these two lines (p = 0.26). Intercepts of them were” reword “these two lines (p = 0.26), the intercepts were”

329-330 “higher in the former than in the latter.” Reword “higher in jellyfish than in fish.”

334 Figure 4 legend requires the same amount of detail as Figure 3 legend.

345 spell out species name when at the start of a sentence

359 It is uncommon to head a section with a question, consider rewording, and line 420.

512 psychology is not the right word, I think its physiology.

358-515 This discussion is well written and well thought out, however I would like to suggest another point as to the positive results of the eDNA compared with the visual census, which is the possibility of polyp stage jellyfish, ephyra, planula, and larval fish stages. This would add to your argument near line 440, 453, 463, 479 and flesh out your conclusion on life stage line 524. I think it would be well worth considering these small stages throughout this discussion.

545 references. I recommend spelling out the journal names in full throughout the reference list, this applies to almost every reference throughout. Also, PLOS ONE in all capitals.

545 also, put all species names in italics, lines 577, 586, 592, 611, 660, 683, 709, 738.

774-778 and S1 and S2 Figures. Add more information to the legend or to the figure including sampling locations and dates in the figure legends and species names so that these figures can stand alone.

6. PLOS authors have the option to publish the peer review history of their article (what does this mean?). If published, this will include your full peer review and any attached files.

Reviewer #1: No

Reviewer #2: No

Reviewer #3: No

---

## [Author Response · Author response to Decision Letter 0]

9 Mar 2020

March 2, 2020

Academic Editor, PhD. Ruslan Kalendar,

PLOS ONE

RE: Revised version of the manuscript for PLOS ONE (PONE-D-19-32313R1)

Dear Editor:

I, along with my co-authors would like to re-submit the attached manuscript entitled newly “Comparing the efficiency of open and enclosed filtration systems in environmental DNA quantification for fish and jellyfish” by English editor (entitled formerly “Comparing the efficiency of open and enclosed filtering systems in fish and jellyfish environmental DNA quantification”) as a research article. (‘Response to Reviewers’, ‘Revised Manuscript with Track Changes’, ‘Manuscript’ figures 6; table 1; Supporting Information 6). The paper was co-authored by Masayuki K. Sakata, Toshifumi Minamoto and Reiji Masuda.

The manuscript has been carefully rechecked and appropriate changes have been made according to the reviewer’s suggestion. The responses to their comments have been prepared and are attached herewith.

We thank you and the reviewers for your thoughtful suggestions and insights, which have enriched the manuscript and produced a more balanced and better account of the research. We hope that the revised manuscript is now suitable for publication in your journal.

We would like to make changes to our financial disclosure; JST CREST Grant Number JPMJCR13A2, Japan and JSPS Grant-in-Aid for Scientific Research (B) 19H03031, Japan.

References were cited in all our laboratory protocols of “Materials and Methods”.

We did not perform the sequence in this study. We described all accession numbers of the sequences of related species for specificity test in S1 Table.

I look forward to your reply.

Sincerely,

Sayaka Takahashi

Faculty of Life and Environmental Science, Shimane University

Nishikawatsu-cho 1060, Matsue, Shimane 6908504, Japan

Phone / Fax: +81-852-32-6513, e-mail: tsayaka@life.shimane-u.ac.jp

Responses to the reviewers’ comments and suggestions

Reviewer #1:

Takahashi and colleagues present the results of an experiment comparing the effects of two different extraction methods — Sterivex and Glass-Fiber Filters — on an eDNA study of fish and jellyfish from the field. In particular, they evaluate the suitability of these two filtering methods in studies comparing species-specific qPCR results to visual counts of biomass. Reassuringly, they show the two filtering methods generally agree with respect to species-specific DNA concentration, and they include some speculation about the benefits of each filtering method. This kind of study is important as eDNA methods work themselves out, and the work is generally applicable beyond the specific case of fish/jellyfish assays. However, my main concerns are statistical: given the data in hand, the authors need to ensure they have a good statistical grounding for their analyses, and that their results are presented in a rigorous way. By pooling across species and engaging in simple linear regression when the data seem to call for some kind of hierarchical model, the authors have smoothed over important difficulties in their dataset.

(Response)

Thank you for your constructive comments. We extensively revised the Materials and methods, Results, and Discussion sections, especially we added revised statistical analysis and discussion of treatment with BAC vs. RNAlater.

#R1-1: Line 77-79: distinguish between open and closed filters; define terms

Our reply: As per your suggestion, we added definition of ’open filter’ (requiring handling, a filter funnel and a vacuum pump) and ‘enclosed filter’ (enclosed in a capsule during filtration and DNA extraction) (L 75-77).

#R1-2: Line 88 and elsewhere: lower-case Greek letter (phi); perhaps accidental?

Our reply: To avoid any confusion, we revised “φ” as “pore size” (L 91-92, 156-157, 165, 421).

#R1-3: Lines 134-136: Might the treatment with BAC vs. RNAlater make any difference in the results?

Our reply: We thank the reviewer for raising this important question. Possible effect of difference between BAC and RNAlater has been discussed in the revise manuscript as follows (L 460-474): 

 “Especially, BAC and RNAlater was used for DNA preservation of the GF/F and Sterivex filtrations, respectively. BAC is cationic surfactant and reduces microbial activity (Yamanaka et al. 2017), whereas RNAlater is a stable reagent which deactivates nucleases (Gorokhova 2005). Water samples with BAC are reported to retain more than 92% of fish DNA for 8-h at ambient temperature (Yamanaka et al. 2017), and those with RNAlater are reported to successfully preserve as the same amount of planktonic DNA for over 1 month as freezing preservation (Gorokhova 2005). This study conducted the process from sampling to preservation in a freezer in short time, and samples were transported on ice immediately after the addition of BAC or RNAlater. Therefore, we assume that there was not much difference in the result due to this step. However, it is also reported that fish species detected by metabarcoding analysis using BAC were lower than those stored on ice (Sales et al. 2019), whereas RNAlater yielded a substantial precipitate that inhibited qPCR amplification of fish (Renshaw et al. 2015). RNAlater stored good quality DNA, but not always in high quantities for metabarcoding analysis of marine organisms (Ransome et al. 2017). The difference between BAC and RNAlater can thus be a subject of future study.”

#R1-4: Lines 292-293: Here begin the statistical issues I have questions about. The analysis and results shown in Figure 3 have several problems, I think:

1. The authors are pooling across species in their linear regression. This alone would seem to violate the assumption of independence among data points, but in addition, they have pooled selectively: even if pooling were acceptable, what would be the justification for splitting apart fish and jellyfish? (Individual species are shown in the supplement, but no statistics are shown, and the un-pooled results are not nearly as strong as the pooled results suggest).

2. The hypothesis being tested here is that, for a given species, the concentration of eDNA detected from samples with GFF and Sterivex are correlated. Accordingly, each species must be treated separately (which will then show, for example, that Halichoeres seems to behave quite differently than the other species tested, etc). Or, if not separately, the authors could do a mixed-effects model in which species is treated as a random effect.

3. It is also not clear that the assumption of normality is met, even after log-transformation, but this is probably a much more minor issue.

Our reply: Thank you for your constructive comments.

1. We split apart fish and jellyfish, because ecological and physiological characteristics are substantially different between these groups of animals, and we had expected that such differences would affect eDNA detection. Previous study in our research group also suggested that emission of eDNA in sea nettles [18] is much more than that of jack mackerel [20], although this comparison was not as straightforward as the present study.

2. We revised statistical analyses according to the suggestion. We have used linear mixed effect model (lmer) analysis with a mixed-effects model in which species was treated as a random effect using R (new Fig 3). The analysis including all the species (new Fig 3, L 322-324) was first presented, followed by the comparison between fish and jellyfish (new Fig 4, L 324-335). We also analyzed each species separately (S1 Fig, L 335-339).

3. Actually, we obtained the same results using “glm” and “glmer” as those using “lm” and “lmer”. We wanted the value of R2, so we used “lm”. Normality was not fully met, yet we applied linear mixed effect model assuming its robustness.

#R1-5: Lines 311-315: if the authors are testing the hypothesis that the eDNA concentrations of the upper and lower parts of the same bag are different, they should do that explicitly. Moreover, by my math, there is a HIGHLY significant correlation between upper and lower samples (treating each species as a single data point of (x,y) with x being upper concentration and y being lower concentration, the slope is 0.98 and r-squared 0.98, or a nearly perfect relationship). And in any case, later in the manuscript the authors determine upper and lower not to matter at all, so why report these results?

Our reply: As you mentioned, there is a HIGHLY significant correlation between upper and lower samples. We tried another statistical analysis using “paired t-test” in R, and it turned out to be the same result as the previous analysis. We reported these results because we wanted to discuss the reason why the eDNA amount captured by Sterivex was different from that captured by the GF/F (L 419-426). The result suggested that a potential bias between the upper and lower layers of water sampling bag would be negligible to affect the difference in result of Sterivex (we sampled from the upper layer in a method of this study) and GF/F (we sampled from the lower layer in a method of this study) (L 427-440). We added the following sentence in the revised Discussion (L 435-437): ”In the method of this study, Sterivex samples were filtered from the upper layer and GF/F from lower layer of water sampling bags.” 

#R1-6: Lines 321 - 332 and Figure 4: Same major pooling problem as before, plus:

1. Does the regression not count the zero values? That is what the lines in Figure 4 imply.

2. The plots indicate substantial Sterivex eDNA with no accompanying visual counts. This may be due EITHER to a) much visually-unobserved biomass, OR b) a high false-positive rate. Given the qPCR assay tests, what is the observed false-positive rate? And then, the authors should discuss the observation in this light.

Our reply: Thank you for your constructive comments and valuable suggestion. We have revised the statistical analyses to use linear mixed effect models (lmer) in which species is treated as a random effect (new Fig 5).

For the tendency that Sterivex detect fish more often than visual census, we generally agree with the view-point provided by the reviewer. This is exactly the issue of “biomass estimation by eDNA from seawater” that we are facing. Regarding the explanation for this phenomenon a), there was 48% of visually-unobserved biomass from our results (there was 53% mismatch case including positive only in the visual census) (L 385-389, 521-522). We showed “Potential causes of this could be currents, seasonality, and animal activity, including their life stage” in the Discussion section (L 522-523). This is actually not surprising, since eDNA survey tends to include a wider spatiotemporal information than visual census as was suggested by previous studies [ex. Ref 5]. Regarding explanation b), We took a great care during the experiment and assume that there were essentially no false-positives with qPCR assay about the experiment. For example, the specificity of the primer sets was checked in each species (L 244-270, 314-318), none of the PCR-negative controls or field blank controls were PCR-amplified (L 290-291), and carry-over contamination was effectively avoided (L 166-167, 170, 291-292).

#R1-7: The equivalent GFF result figure is in the supplement (S2, disaggregated into species, as it should be), but this is a core result of the paper, so it should be in the main piece.

Our reply: As per this suggestion and #R1-9, we have included “GF/F results” and “Sterivex results” in new Fig 5 (L 357-359). The equivalent GF/F result figure (new Fig 6B, D) has also been added in the main piece of this study (L 367-370). The other GF/F results have been added in the supplement (S2-3, S2-4 Figs).

#R1-8: Lines 368 - 370: the authors show no data on *total* eDNA collected, so this hypothesis is untestable. The raw qbit (or similar) results would allow them to test the idea that a larger total amount of eDNA results from a smaller pore size.

Our reply: According to this suggestion we tried an additional experiment to measure total eDNA obtained in each method using Qubit (3.0 dsDNA High Sensitivity). The result was consistent with our result in qPCR having higher eDNA concentration in GF/F compared to Sterivex when the total eDNA was ample. There was however a problem such that four of GF/F samples were undetectably low in concentration, most likely due to the damage during the preservation. Therefore, we consider that the data of Qubit analysis might better not to be included in the present manuscript. (L 417-419).

#R1-9: Overall, why not model the concentration of eDNA as a function of biomass (fixed effect) given species and extraction method (random effects)? This would combine all data into a single, clear and more powerful analysis.

Our reply: As per the suggestion, we have revised the statistical analyses to use linear mixed effect model (lmer) in which species was treated as a random effect using R (new Fig 5). We did not use extraction method as a random effect, because the model did not improve when extraction method added to the model (we compared the models by AIC of “anova” in R) (L 301-304, 357-359, 490-493).

Reviewer #2:

The authors compared two extraction methods for eDNA for jellyfish and fish in two sites. The manuscript is well written.

The experimental design and data analysis were both solid.

Discussion is detailed and very organized.

The conclusion is generally solid. Overall this is a well-conducted study.

There are several areas of improvement for the manuscript.

(Response)

Thank you for your constructive comments.

#R2-1: Four possible explanations on method differences were provided. One of the main differences if the addition of BAC in GF/F. The authors relied on one reference in the literature, which clearly showed that BAC significantly affected eDNA extraction efficiency. Therefore it is likely that the observed result difference was from BAC. However, no control experiments were conducted to compare the effects of the addition of BAC in Strerivex or effects of different concentration of BAC on GF/F. At the minimum, the authors should discuss the mechanisms of BAC of DNA preservation on effects on DNA extraction efficiency.

Our reply: We thank the reviewer for raising this important question. Possible effect of difference between BAC and RNAlater has been discussed in the revise manuscript as follows (L 460-474): 

 “Especially, BAC and RNAlater was used for DNA preservation of the GF/F and Sterivex filtrations, respectively. BAC is cationic surfactant and reduces microbial activity (Yamanaka et al. 2017), whereas RNAlater is a stable reagent which deactivates nucleases (Gorokhova 2005). Water samples with BAC are reported to retain more than 92% of fish DNA for 8-h at ambient temperature (Yamanaka et al. 2017), and those with RNAlater are reported to successfully preserve as the same amount of planktonic DNA for over 1 month as freezing preservation (Gorokhova 2005). This study conducted the process from sampling to preservation in a freezer in short time, and samples were transported on ice immediately after the addition of BAC or RNAlater. Therefore, we assume that there was not much difference in the result due to this step. However, it is also reported that fish species detected by metabarcoding analysis using BAC were lower than those stored on ice (Sales et al. 2019), whereas RNAlater yielded a substantial precipitate that inhibited qPCR amplification of fish (Renshaw et al. 2015). RNAlater stored good quality DNA, but not always in high quantities for metabarcoding analysis of marine organisms (Ransome et al. 2017). The difference between BAC and RNAlater can thus be a subject of future study.”

#R2-2: Another possibility on eDNA extraction efficiency is the difference eDNA extraction procedure as shown in Figure 2. They are similar but not the same ans therefore could still contribute to different results. This should be discussed as well.

Our reply: Thank you for your insightful thoughts. We had pointed out the difference of eDNA extraction procedure in the Discussion sentence (one of four possible explanations: iii) extraction losses) in the previous manuscript, but it seemed to be insufficient. We added a sentence to explain in more detail as follows (L 458-460): “Procedure of eDNA extraction using the GF/F and Sterivex filtration method was similar but not exactly the same (Fig 2), therefore it could still contribute to different results.” 

#R2-3: Please provide melting curve and amplification efficiency of PCR analyses.

Our reply: We used TaqMan probe method, in which the specificity was confirmed. Melting curve could not be created in TaqMan method. Amplification efficiencies of PCR analyses were shown in S2 Table.

Reviewer #3:

This manuscript quantifies the efficiency and capacity of two eDNA analysis techniques and compares them to existing observational biomass census techniques. Their results indicate that there is a positive correlation between the two tested eDNA sampling methods, GF/F and Sterivex, and these two methods were also positively correlated with an underwater visual census. This information is extremely valuable in this new era eDNA research and method development. Overall I found this manuscript very well written and well presented, and I believe this manuscript has strong merit for publication with some reasonable minor changes/edits and some inclusions of finer detail in places. I have provided specific comments and suggestions below, however, please do not feel discouraged by the number of them or how they may be worded, this is in my opinion a very good manuscript.

The number refers to the line number in the original manuscript, followed by the comment.

(Response)

Thank you for your constructive comments. We are very glad that our manuscript improved by lots of your comments and suggestions. We extensively revised the manuscript.

#R3-1: 18 Strange font

Our reply: Thank you for bringing this to our attention. We revised this font (L 18).

#R3-2: 41-42 “more simple” reword. Possibly “more efficient” or similar

Our reply: As per your suggestion, we revised “more simple” as “more efficient” (L 42-43).

#R3-3: 42-43 “After Ficetola [3] applied this method to detect bullfrogs in ponds” consider rewording so that ‘this’ is not the subject of the sentence. “After Ficetola [3] applied a newly developed eDNA method to detect bullfrogs in ponds” or similar.

Our reply: As per your suggestion, we revised “After Ficetola [3] applied this method to detect bullfrogs in ponds” as “After Ficetola [3] applied a newly developed eDNA method to detect bullfrogs in ponds” (L 43-44)

#R3-4: 52 “and tanks [11-12]. eDNA” usually try not to use abbreviations at the beginning of a sentence

Our reply: As per your suggestion, we revised “and tanks [11-12]. eDNA concentration” as “and tanks [11-12]. Concentration of eDNA” (L 54-55).

#R3-5: 57 “visual observation on shore or board [18],” reword this, suggest “visual observations detected via land or vessel based surveys”

Our reply: As per your suggestion, we revised “visual observation on shore or board [18],” as “visual observations via land or vessel based surveys” (L 58-59).

#R3-6: 60 “and mark-recapture techniques” not quite the right word, try “and during mark-recapture experiments”

Our reply: As per your suggestion, we revised “and mark-recapture techniques” as “and during mark-recapture experiments” (L 62).

#R3-7: 67 “species, and this is problematic.” Change to “species, which is problematic.”

Our reply: As per your suggestion, we revised “species, and this is problematic.” as “species, which is problematic.” (L 69).

#R3-8: 67-70 Reword this sentence so that it does not include “we”. Typically write in the third person and not with “we” as the subject of the sentence. Try something like “These studies elucidate the necessity to choose suitable filters”

Our reply: As per your suggestion, we deleted “we” and revised this sentence as “These studies elucidate the necessity to choose suitable filters” (L 69-70).

#R3-9: 86 Similar comment as above, instead of “and we also used” try “which were also used”

Our reply: We used the pore size of 0.7 μm. This was clarified in the revised manuscript as follows: “the most generally used pore sizes were 0.45 μm and 0.7 μm [26], the latter being used in this study.” (L 88-89).

#R3-10: 86 Where possible, try to avoid starting a sentence with an abbreviation (eDNA)

Our reply: As per your suggestion, we moved “in the water” to start of the sentence to avoid starting a sentence with an abbreviation (eDNA) (L 89).

#R3-11: 90-92 It is uncommon to begin a paragraph with a question. Consider changing to “To determine if there are any differences” and edit the sentence to suit.

Our reply: To avoid beginning a paragraph with a question, we changed this paragraph to simpler one (L 94-96).

#R3-12: 94 change to third person rather than use “us” as the subject. Try “will facilitate the ability to estimate” or similar.

Our reply: To avoid using “us”, we changed this paragraph to simpler one (L 97-99).

#R3-13: 96-100 There was the additional aim whereby the upper layer and the lower layer of a sample was compared. I suggest adding it into the aims.

Our reply: Thank you for your valuable suggestion. We added the additional aim (To check for a potential bias between the upper and lower layers of sampling bags, the eDNA concentrations of these two layers were also compared.) (L 102-103).

#R3-14: 109 a space is required at the end of the sentence

Our reply: Thank you for bringing this to our attention. We added a space at the end of the sentence (L 114).

#R3-15: 121 change “Seawater samples, of volume 3 L each” to “Seawater samples (3 L each)”

Our reply: As per your suggestion, we revised “Seawater samples, of volume 3 L each” as “Seawater samples (3 L each)” (L 125).

#R3-16: 121 please explain what a Lamizip is on the first mention, also, is this an adaptation of an existing method of water sample collection? If so please site it.

Our reply: Thank you for your kind suggestion. We added the explanation and model number (Standup Nylon Bag with Zipper and LZ-14) (L 125-126).

#R3-17: 124 how were these water quality quantified. For example, was the salinity measured with a probe or refractometer, also, it is more common to report salinity in parts per thousand (ppt or ‰), practical salinity units (psu). Also, how was visibility measured; visually estimated, or with a secchi disk?

Our reply: Salinity was measured using a conductivity meter with a probe (ES-71, Horiba). The salinity unit with which we measured was so called psu (practical salinity unit) but in our understanding measurement by this apparatus has no unit. Visibility was estimated by a diver who sampled water (Masuda), depth was measured by a SUUNTO D6 diving computer. These descriptions have been included in the revised manuscript (L 128-133, 178-180, 183-185).

#R3-18: 127 This figure and/or figure legend requires more detail and I am not sure how relevant the picture of dead jellyfish is to the sample site. If it is retained, explain the importance of the black arrow, add a scale bar, and label the jellyfish as most people will not know what they are looking at. Please label each of the sample sites to match the text and supplementary material (Otomi and Nagahama), explain what P1 to P6 are, explain the grey curved arrow, describe what L1-L12 are. Each figure should be able to stand alone, I mean this in the nicest way. These suggestions apply to line 127 and to figure 1.

Our reply: Thank you for bringing this to our attention. We added a scale bar and labels of jellyfish in Fig 1(B). We also added explanation of the black arrow, and of the sample points (P1-P6) and visual census lines (L1-12) in figure legend, to match the text and supplementary material (Fig 1, L 135-140).

#R3-19: 130 This figure legend could also use a bit more detail. Consider adding text to clarify the abbreviations, and consider briefly describing each step in the legend. Each figure should be able to stand alone. I am also not sure what the top half of the diagram is there for either, other than looking nice, there is no other information than a diver conducted a visual survey during sample collection.

Our reply: Thank you for your valuable suggestion. We added more detailed description in the procedure, including abbreviations in the legend, and revised this figure (Fig 2, L 142-150). The top half of the diagram is not only to show a visual image of survey during sample collection, but also to show how each species of fish and jellyfish behave in the sea. To obtain these, I took a SCUBA diving license, dived the survey sites, and drew as I have recognized. During the preliminary survey by diving myself, I realized lifestyles (body size, schooling size, activity, pelagic or demersal and so on) of target species are substantially different and considered that eDNA concentration could be affected by lifestyle of each species. I would like to show it visually, and it is the top half of the diagram. We added three “Fig 2” in the Discussion section (L 533, 536, 555) to make use of this figure. Potential difference of eDNA detection depending on life style such as pelagic or demersal is also included referring this figure in the revised manuscript.

#R3-20: 133-137 I would imagine that many of these steps are part of a previously published method. Please state this and site the relevant sources. This will give the methods more credibility. If these are new methods, please explain the importance of each step and/or why it was included.

Our reply: As per your suggestion, we cited the relevant sources “the Environmental DNA Sampling and Experiment Manual (Version 2.1) [45]” (L 153, 212-213).

#R3-21: 155 “salinity in those piers” change to “salinity near each pier”

Our reply: As per your suggestion and next suggestion, we revised “salinity in those piers” as “salinity and depth near pier 1 and pier 2” (L 178).

#R3-22: 156-157 explain which value was associated with which pier (1or2). Explain how the values were measured (as previously suggested) (see comment line 124)

Our reply: We revised explanation of piers in which the water temperature, salinity and depth were measured. The visibility was low near the surface and high near the bottom (or around the sampling points) in both sites. This has been clarified in the revised manuscript. These values were measured in the same manner as the Otomi site presented above (L 128-133) (L 178-180).

#R3-23: 158 change to “in the same manner as the Otomi site presented above.”

Our reply: As per your suggestion, we change this sentence to “in the same manner as at the Otomi site” (L 177).

#R3-24: 161 similar comment for salinity units, and which pier was associated with these measurements, as above there were individual values for each pier.

Our reply: We revised explanation of piers in which the water temperature, salinity was measured. The visibility measurement has been described in more detail (L 183-185).

#R3-25: 171-177 I think the limits of the visual censuses should be included so that the minimum animal size that was detected. For example, no fish were detected under 40 mm in length or jellyfish under 50 mm, even though small specimens may be present.

Our reply: In our routine visual survey the minimum size recorded is 1 cm, yet in the present study the smallest individual recorded were 3 cm. This has been clarified in the revised manuscript (L 200-202).

#R3-26: 205-206 Try to avoid “The procedure that followed was the same as the above.” and briefly explain the steps as there are a lot of procedures above.

Our reply: Thank you for bringing this to our attention. We revised “The procedure that followed was the same as the above” as “Subsequently, we followed the manufacturer’s instructions and eluted in a 100 μL AE buffer before preserving at -20 °C” (L 233-235).

#R3-27: 218 “as being able to amplify specifically each target” suggest reword to amplify each specific target”

Our reply: As per your suggestion, we revised “as being able to amplify specifically each target” as “amplify each specific target” (L 247).

#R3-28: 235 “provided from local fish market.” Reword “provided from a local fish market.”

Our reply: As per your suggestion, we revised “provided from local fish market.” as “provided from a local fish market.” (L 264-265).

#R3-29: 243 consider including more information in the table legend that briefly describes the contents of the table

Our reply: As per your suggestion, we revised the title of Table 1 as “Sequences of primers and probes used for detecting eDNA of five fish and two cnidarian species targeted in this study” (L 272-273).

#R3-30: 252-262 are any of these existing and/or published methods? If so, I suggest the relevant sources are cited which will also add credibility to performing the PCR analysis.

Our reply: As per your suggestion, we added the reference [20] (L 276).

#R3-31: 273 Excel is not considered the best program for statistical analysis and it may be worth checking the results in another program, such as R.

Our reply: Thank you for your valuable suggestion. We checked the results by R. We found slight difference of jellyfish (new Fig 4) and Eja (new S1 Fig) results between Excel and R (concerning rounding off), and we revised them (L 306, 325).

#R3-32: 275 consider rewording the sentence so that it does not start with “S1”

Our reply: In the revised manuscript, this sentence has been deleted.

#R3-33: 285 consider writing in the third person and avoid “our primers” as the sentence subject.

Our reply: As per your suggestion, we changed “our primers” to “primers” (L 317).

#R3-34: 295 reword to “higher than 0”

Our reply: As per your suggestion, we revised “higher from 0” as “higher than 0” (L 327).

#R3-35: 314 please add labels to these values similar to what is above on line 312-313

Our reply: In the revised manuscript, this sentence has been deleted.

#R3-36: 325 “these two lines (p = 0.26). Intercepts of them were” reword “these two lines (p = 0.26), the intercepts were”

Our reply: As per your suggestion, we revised “these two lines (p = 0.26). Intercepts of them were” as “these two lines (p = 0.26), whereas the intercepts were” (L 362).

#R3-37: 329-330 “higher in the former than in the latter.” Reword “higher in jellyfish than in fish.”

Our reply: As per your suggestion, we revised “higher in the former than in the latter.” as “higher in jellyfish than in fish.” (L 366-367).

#R3-38: 334 Figure 4 legend requires the same amount of detail as Figure 3 legend.

Our reply: We revised Figure 4 as new “Fig 6” and added the same amount of detail as Figure 3 (new ”Fig 4”) legend (L 378-383).

#R3-39: 345 spell out species name when at the start of a sentence

Our reply: As per your suggestion, we spelled out “Aurelia” (L 392).

#R3-40: 359 It is uncommon to head a section with a question, consider rewording, and line 420.

Our reply: As per your suggestion, we changed “Which is superior, Sterivex or GF/F?” to “Comparison of performance between GF/F and Sterivex” (L 406).

We also changed “Is it possible Accuracy to estimate biomass of marine species by eDNA?” to “Accuracy of estimating biomass of marine species by eDNA” (L 486).

#R3-41: 512 psychology is not the right word, I think its physiology.

Our reply: We revised the term according to the suggestion (L 581).

#R3-42: 358-515 This discussion is well written and well thought out, however I would like to suggest another point as to the positive results of the eDNA compared with the visual census, which is the possibility of polyp stage jellyfish, ephyra, planula, and larval fish stages. This would add to your argument near line 440, 453, 463, 479 and flesh out your conclusion on life stage line 524. I think it would be well worth considering these small stages throughout this discussion.

Our reply: Thank you for your kind comment and valuable suggestion. We are in agreement with your view. We have added the possibility of eDNA emission from various stage of fish and jellyfish that would not have been visually detected in our argument (L 511-513, 522-523, 533-535, 548-550, 591-592).

#R3-43: 545 references. I recommend spelling out the journal names in full throughout the reference list, this applies to almost every reference throughout. Also, PLOS ONE in all capitals.

Our reply: According to the Submission Guidelines of PLOS ONE, the journal names should be abbreviated, and journal name abbreviations should be those found in the National Center for Biotechnology Information (NCBI) databases. 

https://journals.plos.org/plosone/s/submission-guidelines#loc-references.

In the databases, NLM Title Abbreviation of PLOS ONE is “PLoS One”.

#R3-44: 545 also, put all species names in italics, lines 577, 586, 592, 611, 660, 683, 709, 738.

Our reply: As per your suggestion, we put all species names in italics (L 646-647, 661, 680, 729, 756, 796, 824). “Sakhalin taimen” is not a species name in italics (L 655).

#R3-45: 774-778 and S1 and S2 Figures. Add more information to the legend or to the figure including sampling locations and dates in the figure legends and species names so that these figures can stand alone.

Our reply: As per your suggestion, we showed sampling locations, dates and species names in the figure or the figure legends in S1 and S2 Figures (new “S1 and S2 Figs”) (L 860-864, 866-870, 872-875, 881-884).

---

## [Decision Letter · Decision Letter 1]

19 Mar 2020

PONE-D-19-32313R1

Comparing the efficiency of open and enclosed filtration systems in environmental DNA quantification for fish and jellyfish

PLOS ONE

Dear Dr. Takahashi,

Thank you for submitting your manuscript to PLOS ONE. After careful consideration, we feel that it has merit but does not fully meet PLOS ONE’s publication criteria as it currently stands. Therefore, we invite you to submit a revised version of the manuscript that addresses the points raised during the review process.

Authors need to make minor corrections in accordance with the comments of reviewer #3 before the manuscript is accepted.

We would appreciate receiving your revised manuscript by May 03 2020 11:59PM. To enhance the reproducibility of your results, we recommend that if applicable you deposit your laboratory protocols in protocols.io, where a protocol can be assigned its own identifier (DOI) such that it can be cited independently in the future. For instructions see: http://journals.plos.org/plosone/s/submission-guidelines#loc-laboratory-protocols

We look forward to receiving your revised manuscript.

Kind regards,

Ruslan Kalendar, PhD

Academic Editor

PLOS ONE

Reviewers' comments:

Reviewer's Responses to Questions

**Comments to the Author**

1. If the authors have adequately addressed your comments raised in a previous round of review and you feel that this manuscript is now acceptable for publication, you may indicate that here to bypass the “Comments to the Author” section, enter your conflict of interest statement in the “Confidential to Editor” section, and submit your "Accept" recommendation.

Reviewer #2: All comments have been addressed

Reviewer #3: All comments have been addressed

2. Is the manuscript technically sound, and do the data support the conclusions?

 Reviewer #2: Yes

Reviewer #3: Yes

3. Has the statistical analysis been performed appropriately and rigorously? 

Reviewer #2: Yes

Reviewer #3: Yes

4. Have the authors made all data underlying the findings in their manuscript fully available?

Reviewer #2: Yes

Reviewer #3: Yes

5. Is the manuscript presented in an intelligible fashion and written in standard English?

Reviewer #2: Yes

Reviewer #3: Yes

6. Review Comments to the Author

Reviewer #3: 

Thank you for allowing me to re-review your manuscript, I feel that it is much improved. I have compiled a list of suggestions and comments below, however I will leave these suggestions up to the authors and/or editor to implement at your discretion as most of the comments are wording/style suggestions and small mistakes, all of which should be very easily remedied. The line numbers presented below refer to the line numbers on the revised manuscript.

Line 34 “was relatively low. Concentration of eDNA correlated” insert ‘The’ at the beginning of the sentence “was relatively low. The concentration of eDNA correlated”

Line 54-55 “and tanks [11, 12]. Concentration of eDNA is also positively correlated with” reword by inserting The and change is to was. “and tanks [11, 12]. The concentration of eDNA was also positively correlated with”

Line 81-82 “when using an enclosed, as opposed to an open, filter to detect fish species in ponds [39].” Reword and repunctuate to “when using an enclosed filter, as opposed to an open filter, to detect fish species in ponds [39].”

Line 85-86 “Especially in the eDNA metabarcoding method that utilizes MiFish PCR primers, the number of species” Reword “This may be particularly true when using the eDNA metabarcoding method that utilizes MiFish PCR primers, whereby the number of species”

Line 97-98 “filtration methods will clarify such differences between them. Based on such knowledge, adaptive usage” suggested reword so that “such” is not used twice in a row.

Line 131 If you are going to go with the argument of reporting salinity as unit-less, and you are using a conductivity probe, you still need to state what you are reporting in the methods. I suggest “salinity (measured by a water quality meter with a conductivity probe and reported in practical salinity units (psu): LAQUAact ES-71, Horiba” and then the values of salinity reported throughout the document do not require units.

Line 143-145 I suggest adding a little more detail to explain the inclusion of the picture and I agree with the response you provided in your rebuttal that it should be included. I suggest rewording this “filtration using GF/F and Sterivex. Illustration depicted two species of jellyfish (Aau and Cpa), pelagic fish (Eja and Tja) and demersal fish (Ofa, Hte and Asc). Seawater samples,” and adding this information “filtration using GF/F and Sterivex. The illustration depicts two species of jellyfish (Aau and Cpa), pelagic fish (Eja and Tja) and demersal fish (Ofa, Hte and Asc) and highlights a visual representation of the position in the water column and schooling behaviour of some of these species assemblages during a typical transect. Seawater samples,” or any similar information as this will point out the importance of this illustration.

Line 184 Check this salinity value (2.80) as this would be freshwater “respectively. Salinity near pier 2 was 2.80, and visibility was about 1 m near the surface” I think it is 28.0.

Line 370 change “ware” to “were”

Paragraph Line 452-477 is not presented correctly, has many minor wording mistakes, and the section that was added Line 458-474 does not fit well. I will try and rework a suggestion below, however this is only a suggestion.

Regarding explanation iii), extraction losses may offer a possible explanation as to why the eDNA concentrations obtained by the two filtration methods were different. It is known that the size distribution of eDNA particles varies with environmental conditions [40]. The amount of detected eDNA depends on different eDNA characteristics (size, spatial structure, extra- and intracellular, and particle-bound and free) [30], indicating that the optimal filter choice varies for different extraction methods and there are different protocol combinations suitable for different organisms [28, 58]. At the time of extraction, it is necessary for DNA to flow out of cells, and be removed from filter paper by elution and centrifugation, so that it does not remain on the walls of the tube and column, resulting in eDNA loss.

Furthermore, the procedure of eDNA extraction using the GF/F and Sterivex filtration method was similar, but not exactly the same, during the experiments presented here, which was a necessity to compare the two methods (Fig 2). Therefore it possible that these differences in methods may contribute to differences detected in the results. Specifically, BAC was used for DNA preservation of the GF/F samples, and RNAlater was used for DNA preservation of the Sterivex filtrations. BAC is cationic surfactant and reduces microbial activity [59], whereas RNAlater is a stable reagent which deactivates 463 nucleases [60]. Water samples with BAC are reported to retain more than 92% of fish DNA for 8-h at ambient temperature [59], and those with RNAlater are reported to successfully preserve the same amount of planktonic DNA for over 1 month at ambient temperature as frozen samples [60]. This study conducted the process from sampling to preservation in a freezer in less than seven hours, and the samples were transported on ice immediately after the addition of BAC or RNAlater. Therefore, we assume that there was not much difference in the result due to the preservation step. However, it is also reported that fish species detected by metabarcoding analysis using BAC were lower than those stored on ice [61], whereas RNAlater yielded a substantial precipitate that inhibited qPCR amplification of fish [62]. RNAlater has been shown to store good quality DNA, but not always in high enough quantities for metabarcoding analysis of marine organisms [63]. Therefore, the difference between BAC and RNAlater preservation may be a subject that requires further investigation.

Line 488 “seawater fish” should be changed to “marine fish”

I sincerely hope these suggestions are helpful, and please implement them at your discretion, especially the suggested paragraph edit above.

7. PLOS authors have the option to publish the peer review history of their article (what does this mean?). If published, this will include your full peer review and any attached files.

Reviewer #2: No

Reviewer #3: No

---

## [Author Response · Author response to Decision Letter 1]

23 Mar 2020

March 23, 2020

Academic Editor, PhD. Ruslan Kalendar,

PLOS ONE

Dear Editor:

I, along with my co-authors would like to re-submit the attached manuscript entitled “Comparing the efficiency of open and enclosed filtration systems in environmental DNA quantification for fish and jellyfish” as a research article. (‘Response to Reviewers’, ‘Revised Manuscript with Track Changes’, ‘Manuscript’ figures 6; table 1; Supporting Information 6). The paper was co-authored by Masayuki K. Sakata, Toshifumi Minamoto and Reiji Masuda.

The manuscript has been carefully rechecked and revised in accordance with the comments of reviewer #3. The responses to the comments have been prepared and are attached herewith.

We thank you and the reviewer for your kind suggestions. The manuscript has much more improved. We hope that the revised manuscript is now suitable for publication in your journal.

I look forward to your reply.

Sincerely,

Sayaka Takahashi

Faculty of Life and Environmental Science, Shimane University

Nishikawatsu-cho 1060, Matsue, Shimane 6908504, Japan

Phone / Fax: +81-852-32-6513, e-mail: tsayaka@life.shimane-u.ac.jp

6. Review Comments to the Author

Reviewer #3: 

Thank you for allowing me to re-review your manuscript, I feel that it is much improved. I have compiled a list of suggestions and comments below, however I will leave these suggestions up to the authors and/or editor to implement at your discretion as most of the comments are wording/style suggestions and small mistakes, all of which should be very easily remedied. The line numbers presented below refer to the line numbers on the revised manuscript.

(Response)

Thank you for your constructive comments. We revised our manuscript as per your suggestions. We consider that it was substantially improved.

Line 34 “was relatively low. Concentration of eDNA correlated” insert ‘The’ at the beginning of the sentence “was relatively low. The concentration of eDNA correlated”

Our reply: As per your suggestion, we inserted ‘The’ at the beginning of the sentence (Line 34).

Line 54-55 “and tanks [11, 12]. Concentration of eDNA is also positively correlated with” reword by inserting The and change is to was. “and tanks [11, 12]. The concentration of eDNA was also positively correlated with”

Our reply: As per your suggestion, we revised “and tanks [11, 12]. Concentration of eDNA is also positively correlated with” as “and tanks [11, 12]. The concentration of eDNA was also positively correlated with” (Line 54-55).

Line 81-82 “when using an enclosed, as opposed to an open, filter to detect fish species in ponds [39].” Reword and repunctuate to “when using an enclosed filter, as opposed to an open filter, to detect fish species in ponds [39].”

Our reply: As per your suggestion, we revised “when using an enclosed, as opposed to an open, filter to detect fish species in ponds [39].” as “when using an enclosed filter, as opposed to an open filter, to detect fish species in ponds [39].” (Line 81-82).

Line 85-86 “Especially in the eDNA metabarcoding method that utilizes MiFish PCR primers, the number of species” Reword “This may be particularly true when using the eDNA metabarcoding method that utilizes MiFish PCR primers, whereby the number of species”

Our reply: As per your suggestion, we revised “Especially in the eDNA metabarcoding method that utilizes MiFish PCR primers, the number of species” as “This may be particularly true when using the eDNA metabarcoding method that utilizes MiFish PCR primers, whereby the number of species” (Line 82-84).

Line 97-98 “filtration methods will clarify such differences between them. Based on such knowledge, adaptive usage” suggested reword so that “such” is not used twice in a row.

Our reply: As per your suggestion, we removed the first “such” and revised “such differences” as “the differences” (Line 97).

Line 131 If you are going to go with the argument of reporting salinity as unit-less, and you are using a conductivity probe, you still need to state what you are reporting in the methods. I suggest “salinity (measured by a water quality meter with a conductivity probe and reported in practical salinity units (psu): LAQUAact ES-71, Horiba” and then the values of salinity reported throughout the document do not require units.

Our reply: As per your suggestion, we revised “salinity (measured by a water conductivity meter with a probe: LAQUAact ES-71, Horiba” as “salinity (measured by a water quality meter with a conductivity probe and reported in practical salinity units (psu): LAQUAact ES-71, Horiba” (Line 131-132).

Line 143-145 I suggest adding a little more detail to explain the inclusion of the picture and I agree with the response you provided in your rebuttal that it should be included. I suggest rewording this “filtration using GF/F and Sterivex. Illustration depicted two species of jellyfish (Aau and Cpa), pelagic fish (Eja and Tja) and demersal fish (Ofa, Hte and Asc). Seawater samples,” and adding this information “filtration using GF/F and Sterivex. The illustration depicts two species of jellyfish (Aau and Cpa), pelagic fish (Eja and Tja) and demersal fish (Ofa, Hte and Asc) and highlights a visual representation of the position in the water column and schooling behaviour of some of these species assemblages during a typical transect. Seawater samples,” or any similar information as this will point out the importance of this illustration.

Our reply: Thank you for agreeing with us on that. As per your suggestion, we added a little more detail. We revised “filtration using GF/F and Sterivex. Illustration depicted two species of jellyfish (Aau and Cpa), pelagic fish (Eja and Tja) and demersal fish (Ofa, Hte and Asc). Seawater samples,” as “filtration using GF/F and Sterivex. The illustration depicts two species of jellyfish (Aau and Cpa), pelagic fish (Eja and Tja) and demersal fish (Ofa, Hte and Asc) and highlights a visual representation of the position in the water column and schooling behaviour of some of these species assemblages during a typical transect. Seawater samples,” (Line 144-148).

Line 184 Check this salinity value (2.80) as this would be freshwater “respectively. Salinity near pier 2 was 2.80, and visibility was about 1 m near the surface” I think it is 28.0.

Our reply: Thank you for finding our mistake. We revised “2.80” as “28.0” (Line 187).

Line 370 change “ware” to “were”

Our reply: It is also our mistake. Thank you for finding it. We revised “ware” as “were” (Line 373).

Paragraph Line 452-477 is not presented correctly, has many minor wording mistakes, and the section that was added Line 458-474 does not fit well. I will try and rework a suggestion below, however this is only a suggestion.

Regarding explanation iii), extraction losses may offer a possible explanation as to why the eDNA concentrations obtained by the two filtration methods were different. It is known that the size distribution of eDNA particles varies with environmental conditions [40]. The amount of detected eDNA depends on different eDNA characteristics (size, spatial structure, extra- and intracellular, and particle-bound and free) [30], indicating that the optimal filter choice varies for different extraction methods and there are different protocol combinations suitable for different organisms [28, 58]. At the time of extraction, it is necessary for DNA to flow out of cells, and be removed from filter paper by elution and centrifugation, so that it does not remain on the walls of the tube and column, resulting in eDNA loss.

Furthermore, the procedure of eDNA extraction using the GF/F and Sterivex filtration method was similar, but not exactly the same, during the experiments presented here, which was a necessity to compare the two methods (Fig 2). Therefore it possible that these differences in methods may contribute to differences detected in the results. Specifically, BAC was used for DNA preservation of the GF/F samples, and RNAlater was used for DNA preservation of the Sterivex filtrations. BAC is cationic surfactant and reduces microbial activity [59], whereas RNAlater is a stable reagent which deactivates 463 nucleases [60]. Water samples with BAC are reported to retain more than 92% of fish DNA for 8-h at ambient temperature [59], and those with RNAlater are reported to successfully preserve the same amount of planktonic DNA for over 1 month at ambient temperature as frozen samples [60]. This study conducted the process from sampling to preservation in a freezer in less than seven hours, and the samples were transported on ice immediately after the addition of BAC or RNAlater. Therefore, we assume that there was not much difference in the result due to the preservation step. However, it is also reported that fish species detected by metabarcoding analysis using BAC were lower than those stored on ice [61], whereas RNAlater yielded a substantial precipitate that inhibited qPCR amplification of fish [62]. RNAlater has been shown to store good quality DNA, but not always in high enough quantities for metabarcoding analysis of marine organisms [63]. Therefore, the difference between BAC and RNAlater preservation may be a subject that requires further investigation.

Our reply: We are in agreement with your view. We moved the last sentence to the original place, and revised as per your suggestion (Line 455-483).

Line 488 “seawater fish” should be changed to “marine fish”

Our reply: As per your suggestion, we revised “seawater fish” as “marine fish” (Line 494).

I sincerely hope these suggestions are helpful, and please implement them at your discretion, especially the suggested paragraph edit above.

(Response)

Your suggestions were really helpful for us. Thank you so much.

---

## [Editor Report · Decision Letter 2]

31 Mar 2020

Comparing the efficiency of open and enclosed filtration systems in environmental DNA quantification for fish and jellyfish

PONE-D-19-32313R2

Dear Dr. Takahashi,

We are pleased to inform you that your manuscript has been judged scientifically suitable for publication and will be formally accepted for publication once it complies with all outstanding technical requirements.

With kind regards,

Ruslan Kalendar, PhD

Academic Editor

PLOS ONE

---

## [Editor Report · Acceptance letter]

6 Apr 2020

PONE-D-19-32313R2 

Comparing the efficiency of open and enclosed filtration systems in environmental DNA quantification for fish and jellyfish 

Dear Dr. Takahashi:

I am pleased to inform you that your manuscript has been deemed suitable for publication in PLOS ONE. Congratulations! Your manuscript is now with our production department. 

With kind regards,

on behalf of

Dr. Ruslan Kalendar 

Academic Editor

PLOS ONE